# Structural basis of GABARAP-mediated GABA$_A$ receptor trafficking and functions on GABAergic synaptic transmission

Jin Ye [1,6], Guichang Zou[1,6], Ruichi Zhu [2,3], Chao Kong[1], Chenjian Miao[1], Mingjie Zhang [2,3], Jianchao Li[4✉], Wei Xiong[1,5✉] & Chao Wang [1✉]

GABA$_A$ receptors (GABA$_A$Rs) are the primary fast inhibitory ion channels in the central nervous system. Dysfunction of trafficking and localization of GABA$_A$Rs to cell membranes is clinically associated with severe psychiatric disorders in humans. The GABARAP protein is known to support the stability of GABA$_A$Rs in synapses, but the underlying molecular mechanisms remain to be elucidated. Here, we show that GABARAP/GABARAPL1 directly binds to a previously unappreciated region in the γ2 subunit of GABA$_A$R. We demonstrate that GABARAP functions to stabilize GABA$_A$Rs via promoting its trafficking pathway instead of blocking receptor endocytosis. The GABARAPL1–γ2-GABA$_A$R crystal structure reveals the mechanisms underlying the complex formation. We provide evidence showing that phosphorylation of γ2-GABA$_A$R differentially modulate the receptor's binding to GABARAP and the clathrin adaptor protein AP2. Finally, we demonstrate that GABAergic synaptic currents are reduced upon specific blockage of the GABARAP–GABA$_A$R complex formation. Collectively, our results reveal that GABARAP/GABARAPL1, but not other members of the Atg8 family proteins, specifically regulates synaptic localization of GABA$_A$Rs via modulating the trafficking of the receptor.

[1] MOE Key Laboratory for Membraneless Organelles & Cellular Dynamics, Hefei National Laboratory for Physical Sciences at the Microscale, School of Life Sciences, Division of Life Sciences and Medicine, University of Science and Technology of China, 230027 Hefei, P.R. China. [2] Division of Life Science, State Key Laboratory of Molecular Neuroscience, Hong Kong University of Science and Technology, Clear Water Bay, Kowloon, Hong Kong, P.R. China. [3] Center of Systems Biology and Human Health, Institute for Advanced Study, Hong Kong University of Science and Technology, Clear Water Bay, Kowloon, Hong Kong, P.R. China. [4] Division of Cell, Developmental and Integrative Biology, School of Medicine, South China University of Technology, 510006 Guangzhou, P. R. China. [5] Center for Excellence in Brain Science and Intelligence Technology, Chinese Academy of Sciences, 200031 Shanghai, P.R. China. [6] These authors contributed equally: Jin Ye, Guichang Zou. ✉email: lijch@scut.edu.cn; wxiong@ustc.edu.cn; cwangust@ustc.edu.cn

Type-A γ-aminobutyric acid receptors (GABA$_A$Rs) are the principal mediators of rapid inhibitory synaptic transmission in the vertebrate nervous system[1]. GABA$_A$Rs belong to a superfamily of pentameric ligand-gated ion channels (pLGICs), and naturally occurring GABA$_A$Rs are mostly heteropentamers assembled from numerous subunits[2,3]. Ignoring splice variants, we know that humans have six α subunits, three β subunits, three γ subunits, three ρ subunits, and one each for the ε, δ, θ, and π subunits[4]. Each subunit consists of a hydrophilic extracellular N-terminal domain containing a Cys loop, followed by an α-helical TM1–TM4 transmembrane bundle and a large intracellular domain (ICD) located between TM3–TM4 (Fig. 1a)[5]. A large number of isoforms of GABA$_A$Rs have been identified based on variations in subunit composition, and their diverse arrangements are known to contribute to the functional properties of each specific receptor complex. Among them, the major adult isoform is generally accepted to comprise two α1, two β2, and one γ2 subunits[6]. Notably, the γ2 subunit is the most abundant GABA$_A$R subunit in the central nervous system[7] and is known to be a substrate for protein kinases of several signaling networks[8].

GABA$_A$Rs are expressed ubiquitously in neurons. Accordingly, dynamic changes in their expression and function(s) have been implicated in the regulation of virtually all aspects of brain function[9,10]. Moreover, deficits in the functional expression of GABA$_A$Rs have been clinically associated with epilepsy, anxiety disorders, cognitive deficits, schizophrenia, depression, and substance abuse[11–14]. The level of GABA$_A$R accumulation on the neuronal plasma membrane is regulated via two primary mechanisms: the assembly of subunits into transport-competent GABA$_A$Rs (trafficking pathway for these receptors to the plasma membrane), and the endocytosis-mediated recycling and degradation of the GABA$_A$Rs[15,16]. A number of proteins have been functionally demonstrated in GABA$_A$Rs trafficking to the postsynaptic membrane and in the anchoring of receptors onto the cytoskeleton[15].

One such protein, GABA$_A$R-associated protein (GABARAP), is an isoform of the mammalian autophagy-related protein 8 (Atg8) family protein, a ubiquitin-like protein required for the formation of autophagosomal membranes[17]. Previous studies have reported that GABARAP can directly bind to the TM3–TM4 intracellular loop of GABA$_A$ receptor γ2 subunit (γ2-GABA$_A$R) with low affinity[18–20], and the co-expression of GABARAP and GABA$_A$Rs in COS7 cells, quail fibroblasts, oocytes, and cultured hippocampal neurons have been shown to increase the surface levels of GABA$_A$Rs[21–23]. Several studies have proposed that GABARAP may promote translocation of GABA$_A$Rs from intracellular compartments to the somatodendritic plasma membrane, potentially leading to GABARAP-dependent potentiation of inhibitory synapses[24,25]. However, the molecular mechanisms of GABARAP in regulating GABAergic synaptic transmission remain to be elucidated. Notably, no structural characterization on the binding between GABA$_A$Rs and GABARAP is available to date.

In the present study, we first demonstrate that the previously reported octadecapeptide from the ICD of γ2-GABA$_A$R does not bind to GABARAP. Instead, a previously unrecognized fragment of the TM3–TM4 intracellular loop of γ2-GABA$_A$R (γ2-GIM) specifically binds to GABARAP/GABARAPL1, but not other Atg8 family members. The high-resolution crystal structure for the γ2-GIM in complex with GABARAPL1 reveals mechanistic details for the specific interaction between the γ2 subunit and GABARAPL1. Intriguingly, the γ2-GIM also binds to clathrin adaptor protein AP2 with an affinity higher

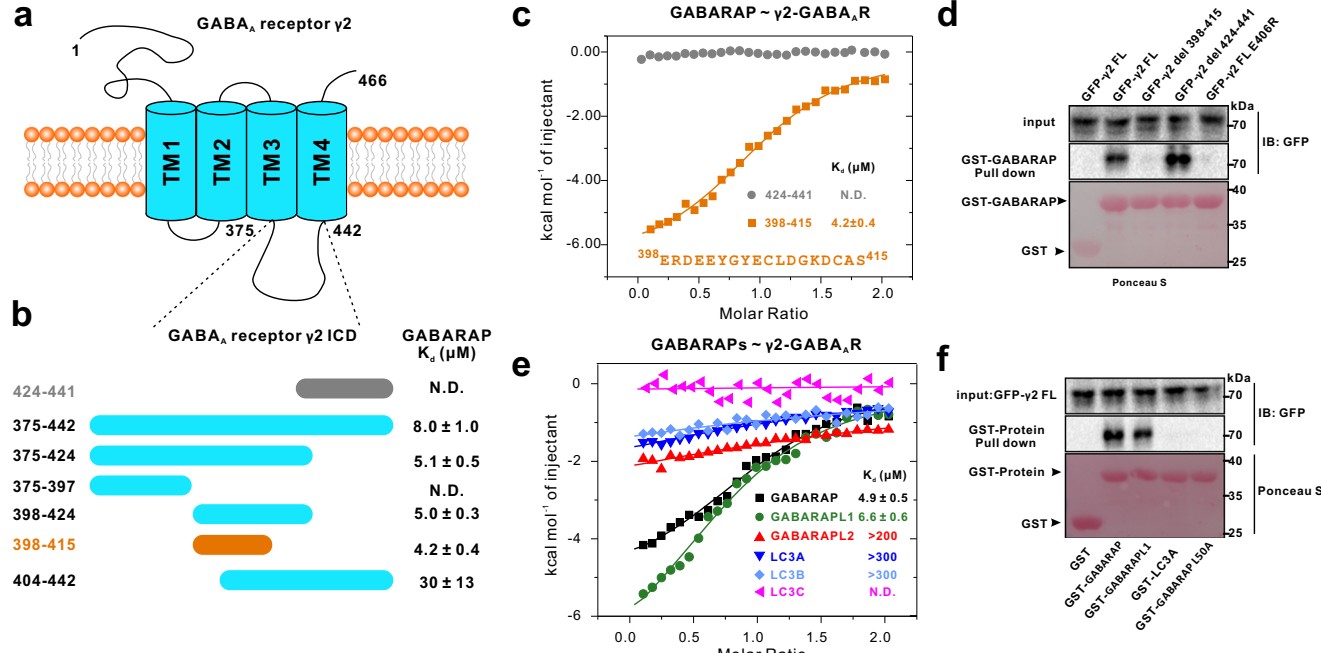

**Fig. 1 Identification of an 18-aa intracellular loop region of the GABA$_A$R γ2 subunit that can bind to GABARAP. a** A schematic diagram showing the domain organization of the GABA$_A$ receptor γ2 subunit. **b** ITC-based mapping of the minimal GABARAP binding region in the TM3–TM4 ICD. The minimal and complete GABARAP-binding region identified is highlighted in orange. The 'N.D.' denotes that these constructs had no detectable binding to GABARAP. **c** ITC-derived binding curves comparing binding affinities of the different truncation variants of the TM3–TM4 ICD for binding with GABARAP. **d** GST-pull down assays showing that the fragment 398-415 (GIM) of γ2-GABA$_A$R is responsible for GABARAP binding; besides, key residue of γ2-GABA$_A$R E406 is required for the GABARAP/γ2-GABA$_A$R interaction. **e** ITC-derived binding curves comparing binding affinities between GABA$_A$R γ2-GIM and 6 Atg8 family proteins. **f** GST-pull down assays showing that full length γ2-GABA$_A$R specifically binds to GABARAP and GABARAPL1 but not LC3A; besides, key residue of GABARAP L50 is required for the GABARAP/γ2-GABA$_A$R interaction.

than to GABARAP. Using whole-cell patch clamp recording of HEK-293 cells expressing various forms of GABARAP and GABA$_A$Rs, we confirmed that GABARAP enhances GABA currents and found that GABARAP functions to increase the membrane levels of GABA$_A$Rs via a trafficking pathway rather than by blocking endocytosis. Finally, we demonstrated that the amplitudes of miniature inhibitory postsynaptic currents (mIPSCs) for GABAergic synapses are reduced upon blocking GABARAP–GABA$_A$R complex formation with potent inhibitory peptides derived from giant ankyrins. In summary, our study provides mechanistic insights into the interactions between GABA$_A$Rs and GABARAP and deepens our understanding of the functional modes of GABARAP in GABAergic synaptic transmission.

## Results

**Identification of an 18-aa intracellular loop region of the GABA$_A$ receptor γ2 subunit that can bind to GABARAP.** Previous studies concluded that the octadecapeptide RTGAWRHGRIHIRIAKMD (residues 424–441) from the TM3–TM4 intracellular loop of the GABA$_A$ receptor γ2 subunit was necessary and sufficient for interacting with GABARAP[19]. However, when we used isothermal titration calorimetry (ITC) assay to characterize the interaction between GABARAP and this octadecapeptide, we found with surprise that there was no binding between GABARAP and the octadecapeptide (Fig. 1a–c). Given these unexpected findings and seeking molecular insights about the specific interaction(s) of GABARAP with γ2-GABA$_A$R, we first examined a sequence alignment of the TM3–TM4 ICD of the GABA$_A$R γ subunits in multiple animals. The ICD of the γ2 subunit is highly conserved among the examined species (Supplementary Fig. 1a), and is clearly more strongly conserved than homologous sequences from other GABA$_A$R γ subunits (γ1–3, Supplementary Fig. 1 lower panel). The undetected binding for the octadecapeptide prompted us to conduct a systematic mapping analysis of the binding of GABARAP with various recombinant segments from different positions of the TM3–TM4 ICD. The dissociation constant ($K_d$) for the full length TM3–TM4 ICD (residues 375–442) binding to GABARAP was ~8 μM (Fig. 1a, b, and Supplementary Fig. 1b). A recombinant of the TM3–TM4 ICD lacking the aforementioned 424–441 octadecapeptide could still bind to GABARAP, doing so with a $K_d$ of ~5.1 μM (Fig. 1b), a finding clearly suggesting that some other region of the TM3–TM4 ICD that can interact with GABARAP.

Further ITC-based mapping showed that an evolutionarily conserved 18-aa intracellular loop region (residues 398–415) that is located next to the 424–441 octadecapeptide is both necessary and sufficient for binding to GABARAP, exhibiting a $K_d$ of ~4.2 μM (Fig. 1b, c). We further showed that purified GST-GABARAP can successfully pull-down full length (FL) GFP-γ2-GABA$_A$R wild-type (WT) protein expressed in mammalian cells (Fig. 1d). In contrast, binding was abolished between the full length receptor with a 398–415 deletion or an E406R variant (see details below for the variant). However, GABARAP can still interact with GABA$_A$R with a 424–441 deletion, a finding indicating that this previously reported motif may not participate in binding (Fig. 1d). Analytical gel filtration chromatography further confirmed the ability of this 398–415 intracellular loop region binding to GABARAP (Supplementary Fig. 2a). Collectively, these results correct previous assumptions about the binding capacity of the 424–441 octadecapeptide and demonstrate that a previously unappreciated 18-aa intracellular loop region of the γ2 subunit mediates the binding of the GABA$_A$R with GABARAP.

**GABA$_A$ receptor γ2 subunit GIM binds to GABARAP/GABARAPL1 but no other Atg8 family members.** GABARAP belongs to the Atg8 family, and the mammalian Atg8 family contains six members that are classified into two subfamilies: GABARAPs (including GABARAP, GABARAPL1, and GABARAPL2) and LC3s (including LC3A, LC3B, and LC3C)[17]. These six proteins share a high degree of sequence similarity and are known to exert some redundant functions[26,27]. We used ITC and analytical gel filtration assays to test whether the 18-aa GABA$_A$R γ2 subunit GIM (GABARAP interacting motif, 398–415; henceforth γ2-GIM) can interact with other Atg8 family members. The ITC analysis showed that γ2-GIM can interact with GABARAPL1 with essentially the same affinity ($K_d$ ~6.6 μM) but does not interact with GABARAPL2 or any of the LC3s (Fig. 1e). Consistently, analytical gel filtration assays also showed that γ2-GIM specifically binds to GABARAP and GABARAPL1 but none of the other Atg8 family members (Supplementary Fig. 2a). We also utilized GST pull-down assays to verify the GABA$_A$R binding specificity with diverse Atg8 members. Our results showed that GST-GABARAP and GST-GABARAPL1 could bind to GFP-γ2-GABA$_A$R FL expressed in HEK-293 cells, whereas no binding was detected for GST-LC3A or a GST-GABARAP L50A variant (Fig. 1f, see details below for the variant).

**The overall structure of GABARAPL1 in complex with γ2-GIM.** To elucidate the mechanistic basis of the interaction between the γ2-GIM and GABARAP, we next attempted to determine their complex structure. Given the high sequence similarity between GABARAP and GABARAPL1 (sequence identity 87.1%, Supplementay Fig. 2b), as well as its similar binding affinity with γ2-GIM ($K_d$: 4.9 vs. 6.6 μM), we tried crystallization of both GABARAP–γ2-GIM and GABARAPL1–γ2-GIM complex. However, GABARAP–γ2-GIM yielded no crystals. We successfully obtained good crystals for a GABARAPL1–γ2-GIM complex, which diffracted to a resolution of 1.95 Å (Supplementary Table 1). The GABARAPL1–γ2-GIM complex structure was solved using the molecular replacement method (Supplementary Table 1).

In the complex, GABARAPL1 adopts a typical architecture for an Atg8 ortholog protein, consisting of 2 N-terminal α-helices and a ubiquitin-like core (Fig. 2a). The clearly defined region of the γ2-GIM in our complex structure contains 13 highly conserved residues (EEYGYECLDGKDC; 401–413) (Fig. 2a, b). Generally, the γ2-GIM appears as an overturned hook lying in an elongated groove of GABARAPL1 formed by the α1-helix, α2-helix, β2-strand, and α3-helix. Thus, our structural data support the plausibility of our systematical biochemical data for the specific binding of γ2-GIM and GABARAPL1.

**Characterization of key residues at the interface of the GABARAPL1–γ2-GIM complex.** A detailed structural analysis of the inter-molecular interface between GABARAPL1 and γ2-GIM in our complex structure revealed a classic GIM-dependent binding mode (Fig. 2c). In particular, the aromatic side chain of γ2-GIM Y405 is deeply inserted into a hydrophobic pocket of GABARAPL1 formed by the hydrophobic side chains of I21, P30, L50, and F104 residues (together with the aliphatic side chain of K48) (Fig. 2c). The hydrophobic side chain of γ2-GIM L408 occupies another hydrophobic pocket of GABARAPL1 that is positioned in the β2/α3 groove and formed by the Y49, V51, F60, and I64 residues of GABARAPL1 (Fig. 2c).

In addition, hydrogen bonds were observed between GABARAPL1 E17 and γ2-GIM Y405 and between GABARAPL1 K66 and

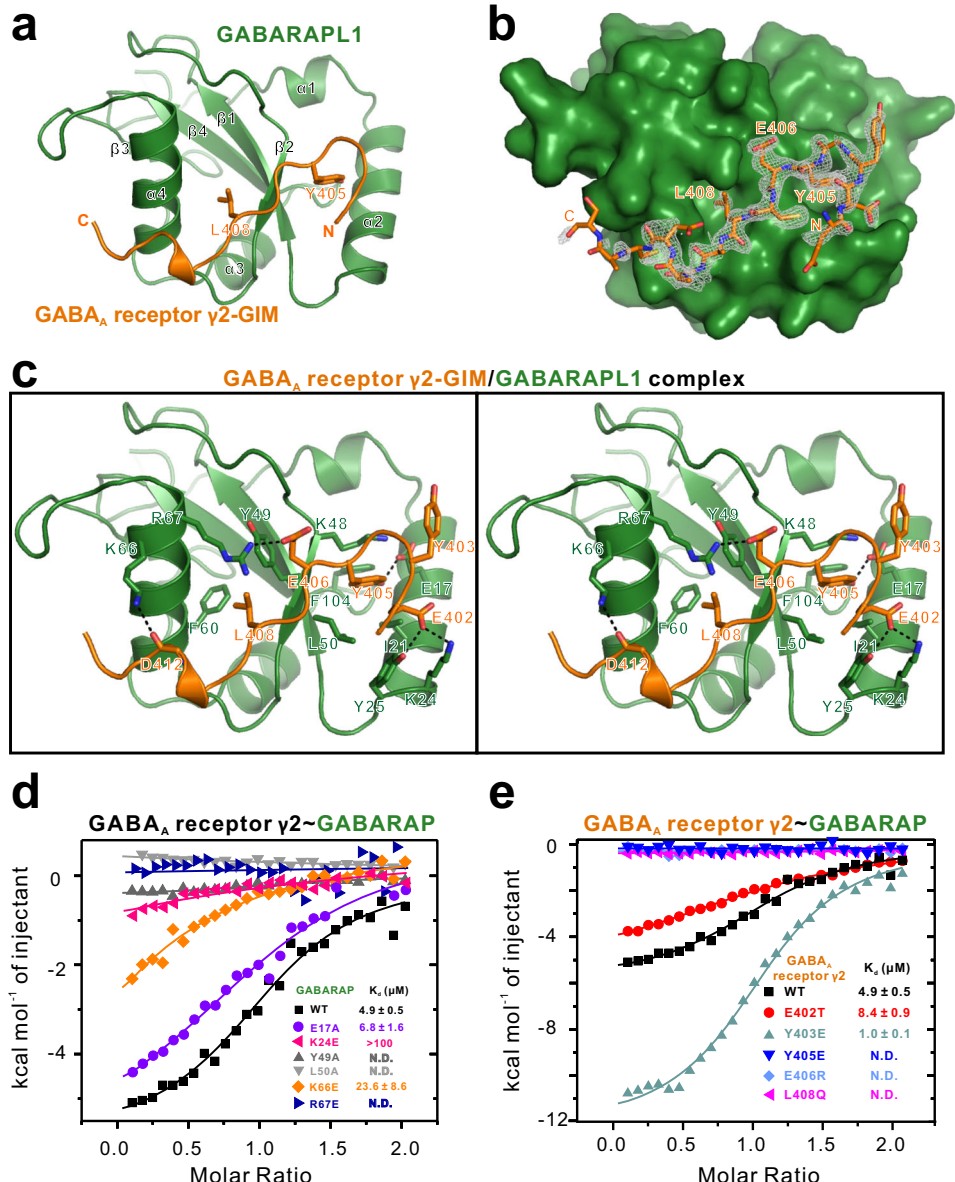

**Fig. 2 Overall structure of GABARAPL1 in complex with γ2-GIM and characterization of key residues at the interface of the GABARAPL1–γ2-GIM complex. a** Ribbon representation model showing the overall structure of the GABARAPL1–γ2-GIM complex. In this drawing, GABARAPL1 is shown in forest green and γ2-GIM is in orange. The coordinates and structure factors of the GABARAP and γ2-GABA$_A$R complex have been deposited in the Protein Data Bank with accession number 7CDB. **b** The $F_o−F_c$ map of γ2-GIM showing that the densities of residues 402–413 can be clearly assigned. **c** Stereo view showing the detailed interactions between γ2-GIM and GABARAPL1. The hydrogen bonds involved in the binding are shown as dotted lines. **d** ITC-derived binding curves comparing the binding affinities between γ2-GIM and GABARAP or its six mutant variants (E17A, K24E, Y49A, L50A, K66E, and R67E). **e** ITC-derived binding curves comparing binding affinities between GABARAP and γ2-GIM or its five mutant variants (E402T, Y403E, Y405E, E406R, and L408Q).

γ2-GIM D412 (Fig. 2c). Our structure also clearly indicated that the GABARAPL1–γ2-GIM complex is further stabilized by two charge–charge interactions: one is located at the N-terminal area of GIM–GABARAPL1 interface and is mediated by the negatively charged E402 of γ2-GIM and the positively charged K24 of GABARAPL1; the other is formed between the negatively charged γ2-GIM residue E406 and the positively charged R67 residue of GABARAPL1 (Fig. 2c). Note that all of these interface residues of γ2-GIM are strictly conserved in the animal species we examined (Supplementary Fig. 1).

We next used quantitative ITC analysis to characterize the functional impacts of the candidate key functional interface residues with a large variety of mutational variants of γ2-GIM

and GABARAP. We observed essentially abolished (Y49A and L50A) binding in assays testing single mutant variants of the GABARAP hydrophobic pocket residues in binding with γ2-GIM (Fig. 2d). For the hydrophilic residues, K24E variant profoundly reduced binding between GABARAP and GIM; R67E totally abolished the interaction; E17A slightly weakened the interaction, and K66E weakened the interaction by around four times (Fig. 2d). For γ2-GIM, we evaluated five mutant variants in binding with GABARAP: the E402T variant (this position is a Thr in human γ3 subunit) slightly weakens the binding (a $K_d$ of 8.4 vs. 4.9 µM for WT); the Y403E variant (phosphomimic mutation) increases the binding affinity to around 5 folds; the Y405E (phosphomimic mutation), E406R (charge reversal to

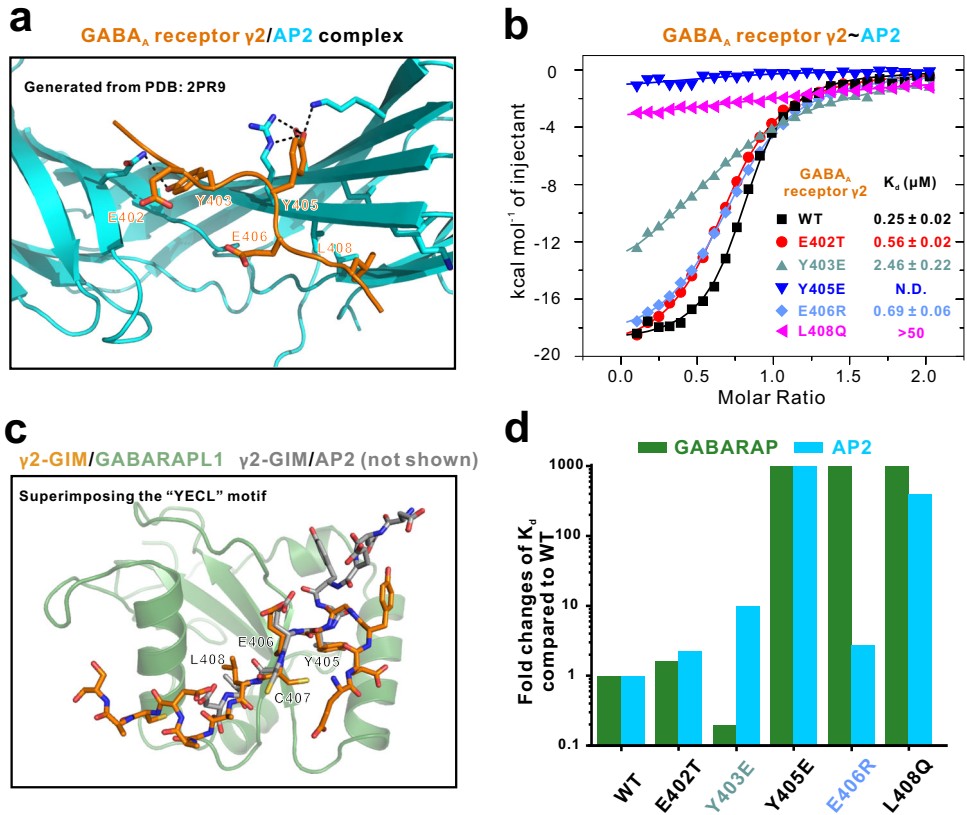

**Fig. 3 Distinct molecular mechanisms regulating γ2-GIM's specific interactions with GABARAP or AP2. a** The ribbon representation model showing the overall structure of the previously reported AP2-γ2-GIM complex. The figure is generated using the previous reported peptide complex structure (PDB code 2PR9[28]). In this drawing, AP2 is shown in cyan and γ2-GIM is shown in orange. **b** ITC-derived binding curves comparing the binding affinities between AP2 and GABA$_A$R γ2-GIM or its five mutant variants (E402T, Y403E, Y405E, E406R, and L408Q). **c** Compare the conformations of γ2-GIM in complex with GABARAP or AP2 by superimposing the "YECL" motif. **d** Bar graph of fold changes of $K_d$ values compared to WT showing the impacts on binding resulting from mutations of key residues in the GABA$_A$R γ2-GIM. Titrations for WT and E406R mutant were performed three times; others were performed once. $K_d$ values are reported in Supplementary Table 2.

introduce a charge repulsion effect), and L408Q (hydrophobic to polar residue) variants totally abolish the binding with GABARAP (Fig. 2e and Supplementary Table 2). We have also performed ITC assays with several of the corresponding GABARAPL1 variants, to further validate our biochemical and structural conclusions. In agreement with our structural observations, K24E, L50A, and R67E from GABARAPL1 all abolished any interaction with γ2-GIM (Supplementary Fig. 3).

**GABARAP and clathrin adaptor protein AP2 compete with each other in binding with γ2-GABA$_A$R via the overlapping motif.** Previous studies have reported that the clathrin adaptor protein AP2—which functions to control the level of GABA$_A$R at the cell surface by regulating the extent of endocytosis—can interact with a motif of the GABA$_A$R γ2 subunit intracellular loop known as the "YECL" motif (Fig. 3a, PDB: 2PR9)[28]. This motif shares 10 residues (DEEYGYECLD; 400–409) overlapping with the γ2-GIM sequence we identified based on our complex structure and binding assays with GABARAP (Figs. 2c and 3a). Given this overlap, we wonder if there is any potential relationships between the interactions of γ2-GABA$_A$R with GABARAP and with AP2. To address this, we first measured the binding affinity of γ2-GIM with AP2 using ITC-based assays: γ2-GIM binds to AP2 with a $K_d$ of ~0.25 μM (Fig. 3b and Supplementary Table 2), an affinity 20-fold stronger than that for GABARAP–γ2-GIM binding (Supplementary Table 2). We further confirmed

this stronger binding affinity of AP2–γ2-GIM complex than GABARAP–γ2-GIM complex through both analytic gel filtration and competitive GST pull down assay, showing that GABARAP cannot compete with equimolar AP2 in binding with γ2-GIM (Supplementary Fig. 4).

We then compared complex structures including our GABAR-APL1–γ2-GIM complex and a previously reported AP2–YECL motif complex by superimposing the YECL motif (Figs. 2c, 3a, c)[28]. Most obviously, there were clear similarities in the spatial arrangement of the γ2-GIM and YECL motif residues at their respective interfaces with GABARAPL1 and AP2. Specifically, the side chains of γ2-GABA$_A$R residues Y405 and L408 each interact with binding pockets present in GABARAPL1 and AP2. Recall that mutation of the Y405 and L408 γ2-GIM residues abolished binding with GABARAP; we also conducted ITC assays using AP2 and the aforementioned γ2-GIM five mutant variants. The Y405E and L408Q γ2-GIM mutant variants, respectively, abolished and attenuated the binding interaction with AP2 (Fig. 3b). Despite these similarities, we also detected clear distinctions in the interactions governing the two complex interfaces (Fig. 3c). For example, we found that mutation of the γ2-GIM E406 to Arg hardly affect the interaction with AP2 but totally abolished GABARAP binding; Y403E γ2-GIM mutant decreases the binding affinity about 10 folds compared to WT in binding to AP2, while the same mutant increases the binding affinity around 5 folds in binding to GABARAP (Fig. 3d and Supplementary Table 2). Thus,

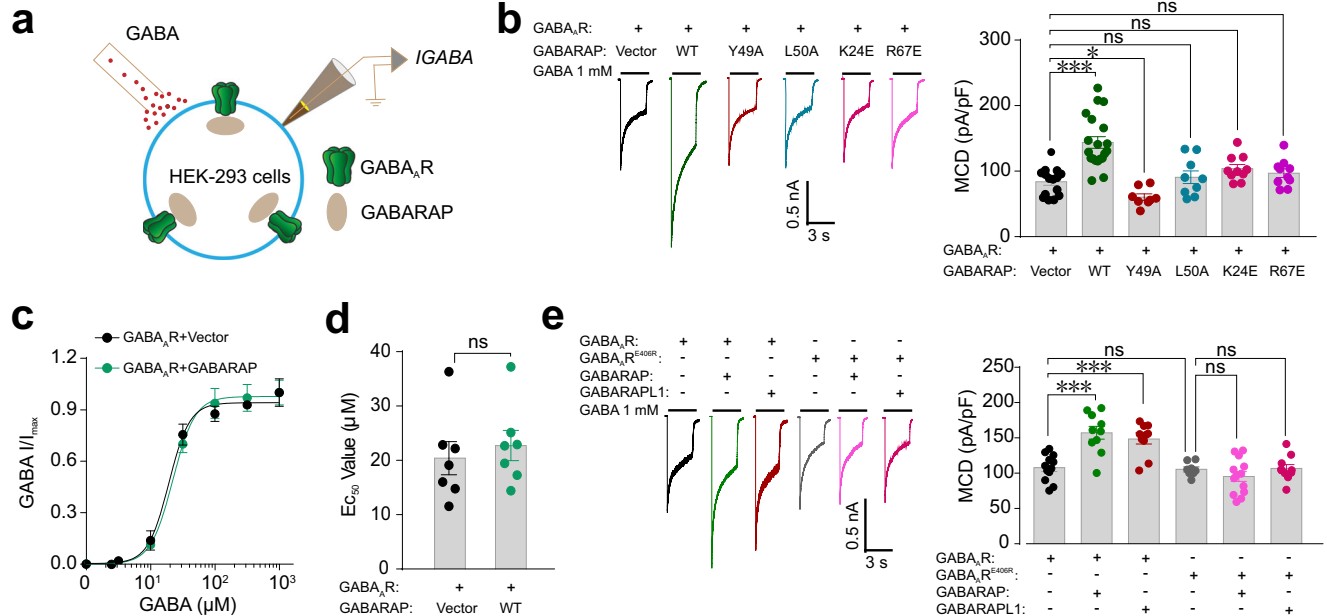

**Fig. 4 Effects of GABARAP on GABA$_A$R-mediated current densities in HEK-293 cells co-expressing diverse GABARAP and GABA$_A$Rs variants.**
**a** Schematic diagram for patch-clamp recordings of HEK-293 cells. **b** Representative trace records and average values of GABA currents ($I_{GABA}$) activated by 1 mM GABA in HEK-293 cells co-expressing GABA$_A$Rs (α1β2γ2) and WT or mutant GABARAP. $n = 15$, 20, 8, 9, 10, and 10 of each group (from left to right). Data are represented as the mean ± SEM. ***$P = 0.0001$, GABA$_A$R vs. GABA$_A$R+GABARAP; *$P = 0.0108$, GABA$_A$R vs. GABA$_A$R+GABARAP$^{Y49A}$ based on one-way ANOVA; ns not significant ($P > 0.05$). **c** Dose–response curves of $I_{GABA}$ in HEK-293 cells co-expressing GABA$_A$Rs (α1β2γ2) and GABARAP. The data were normalized to $I_{max}$ of the each group. GABA at 1, 3, 10, 30, 100, 300, and 1000 μM were selected. Data were fit using the Hill equation with variable slope. Data are represented as the mean ± SEM. $n = 7$. **d** EC$_{50}$ values of $I_{GABA}$ induced by increasing GABA concentrations in HEK-293 cells co-expressing GABA$_A$R (α1β2γ2) and GABARAP. $n = 7$. Data are represented as the mean ± SEM. ns not significant ($P > 0.05$) based on unpaired $t$ test. **e** Representative trace records and average values of GABA currents activated by 1 mM GABA in HEK-293 cells co-expressing GABA$_A$R (α1β2γ2$^{E406R}$) and GABARAP or GABARAPL1. $n = 12$, 10, 13, 8, 10, and 10 of each group (from left to right). Data are represented as the mean ± SEM. ***$P = 0.0001$, GABA$_A$R vs. GABA$_A$R + GABARAP; ***$P = 0.0010$, GABA$_A$R vs. GABA$_A$R + GABARAPL1 based on one-way ANOVA; ns, not significant ($P > 0.05$).

E406R and Y403E γ2-GIM mutant variants could serve as specific tools to distinguish functional impacts of GABA$_A$R in binding with GABARAP vs. AP2 (Fig. 3d and Supplementary Table 2).

**GABARAP enhances GABA$_A$R-mediated current densities in HEK-293 cells.** Given that GABARAP can interact with the γ2-GABA$_A$R directly, it may function in the trafficking of GABA$_A$Rs and therefore affect GABA$_A$R-mediated chloride currents. Pursuing this, we used patch clamp recording to measure GABA$_A$R responses under GABA elicitation in HEK-293 cells co-expressing the full GABA$_A$Rs (α1β2γ2) and GABARAP (Fig. 4a). Presence of GABARAP caused an obvious increase in the GABA-activated maximum current density (MCD) compared to cells expressing the GABA$_A$Rs alone (Fig. 4b). Such increased GABA MCD might relate with elevated levels of GABA$_A$Rs at the surface of the HEK-293 cells, based on previous studies reporting that co-expression of GABARAP and GABA$_A$Rs in COS7 cells, oocytes, and cultured hippocampal neurons increases the surface levels of GABA$_A$Rs[21–23]. In addition, we have also verified the effect of GABARAP on GABA potency by examining concentration–response curves in HEK-293 cells co-expressing GABA$_A$R and GABARAP. Although the presence of GABARAP significantly increased GABA induced maximum current (Fig. 4b), it did not alter the EC$_{50}$ value for GABA (Fig. 4c, d). These findings support that GABARAP does not apparently impact the potency of GABA on GABA$_A$R.

Consistent with our aforementioned ITC-assay results demonstrating that Y49A, L50A, K24E, and R67E GABARAP variants do not bind with γ2-GIM, we detected no increases in GABA MCD when we expressed these GABARAP variants in HEK-293 cells alongside full GABA$_A$Rs (Fig. 4b). Conversely, recording of cells co-expressing WT GABARAP and the binding-deficient γ2-GIM variant E406R revealed a significant reduction in the GABA MCD (Fig. 4e). Furthermore, we showed that GABARAPL1 has similar effect with GABARAP on the GABA currents (Fig. 4e). We have also selected α5-containing GABA$_A$Rs (α5β2γ2) as a representative for an extrasynaptic GABA$_A$R[29,30], and examined the effect of GABARAP on this type of GABA$_A$R. GABARAP also significantly increased the GABA$_A$R-mediated current as induced by 1 mM GABA (Supplementary Fig. 5a), suggesting that γ2 containing extrasynaptic GABA$_A$Rs are also sensitive to regulation by GABARAP. Thus, specific binding between GABARAP/GABARAPL1 and the GABA$_A$R γ2 subunit increases the extent of GABA MCD possibly by promoting accumulation of the GABA$_A$Rs at the cell surface.

To support the above electrophysiological data, we also performed cell surface biotinylation analysis of GABA$_A$R in HEK-293 cells with or without GABARAP WT or mutation variants. Our results showed that both GABARAP and GABAR-APL1 could increase levels of receptors at the membrane surface; moreover, such increases did not occur with binding-deficient mutation variants of GABARAP L50A or R67E or of GABA$_A$R E406R (Supplementary Fig. 5b).

**GABARAP functions to stabilize GABA$_A$Rs via promoting its trafficking pathway.** As mentioned above, the membrane levels of GABA$_A$Rs are known to be modulated through constitutive clathrin adaptor AP2 complex-dependent endocytosis of

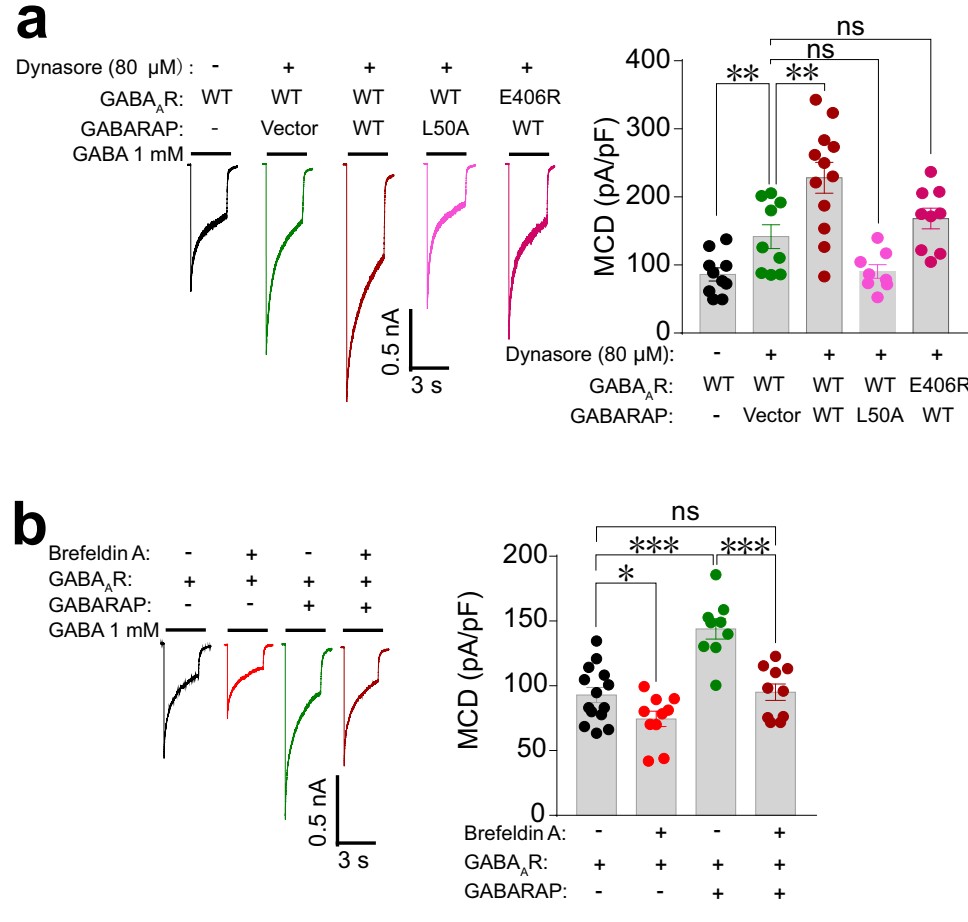

**Fig. 5 GABARAP functions to stabilize GABA$_A$Rs via promoting its trafficking pathway. a** Representative trace records and average values of GABA currents activated by 1 mM GABA in HEK-293 cells co-expressing GABA$_A$R (α1β2γ2) and WT or mutant GABARAP, with or without treatment using the endocytosis inhibitor dynasore (80 μM, 2 h). $n = 10$, 9, 12, 8, and 9 of each group (from left to right). Data are represented as the mean ± SEM. **$P =$ 0.0060, GABA$_A$R vs. GABA$_A$R+Dynasore; **$P =$ 0.0070, GABA$_A$R+Dynasore vs. GABA$_A$R+GABARAP+Dynasore based on one-way ANOVA; ns, not significant ($P > 0.05$). **b** Representative trace records and average values of GABA currents activated by 1 mM GABA in HEK-293 cells co-expressing GABA$_A$R (α1β2γ2) and GABARAP with or without treatment of the protein trafficking inhibitor Brefeldin A (5 μg/mL, 30 min). $n = 14$, 10, 9, and 10 of each group (from left to right). Data are represented as the mean ± SEM. *$P = 0.0431$, GABA$_A$R vs. GABA$_A$R+Brefeldin A; ***$P = 0.0001$, GABA$_A$R vs. GABA$_A$R +GABARAP; ***$P = 0.0001$, GABA$_A$R+GABARAP vs. GABA$_A$R+GABARAP+Brefeldin A based on one-way ANOVA; ns not significant ($P > 0.05$).

GABA$_A$Rs from cell membranes[28]. Given our findings that AP2 and GABARAP bind overlapping regions of the γ2 subunit, we speculated that GABARAP may stabilize the GABA$_A$R either by preventing its removal from the cell surface via AP2-dependent endocytosis or by promoting the trafficking of GABA$_A$Rs from Golgi to cell membranes. Pursuing these two possibilities, we first performed a parallel experiment using HEK-293 cells expressing the full GABA$_A$Rs treated with two concentrations of the endocytosis inhibitor dynasore (80 and 120 μM, for 2 h; this molecule can block endocytosis by more than 90%[31]) to confirm dynasore blocks endocytosis to a sufficient extent. Both concentrations of dynasore treatment dramatically increased GABA MCD (Supplementary Fig. 6). Compared with the 80 μM dynasore, the higher concentration of dynasore did not further elevate the GABA MCD (Supplementary Fig. 6). Interestingly, we detected a further enhancement of GABA MCD when we co-expressed GABARAP and GABA$_A$R in HEK-293 cells with treatment of the same dose of dynasore (80 μM) (Fig. 5a). In contrast, the binding-deficient GABARAP variant L50A caused no further enhancement of GABA MCD in the presence of dynasore (Fig. 5a). Likewise, and consistent with our bio-chemical data, no further enhancement was observed in cells

co-expressing GABARAP and the γ2-GIM E406R variant (Fig. 5a).

To further investigate GABARAP's function in promoting the receptor on the cell membrane, we have also performed experiments that enabled direct evaluation of the effects of GABARAP on receptor trafficking by treatment with Brefeldin A (BFA), which induces Golgi disassembly and is widely used to block the trafficking of membrane proteins[32,33]. Our results showed that BFA treatment (5 μg/mL, 30 min) alone could reduce the GABA current densities, suggesting a decrease in GABA$_A$R membrane localization (Fig. 5b). Further, BFA treatment blocked the GABARAP-mediated increase in GABA currents as induced by 1 mM GABA (Fig. 5b), a finding demonstrating that GABARAP promotes receptor membrane localization via a trafficking pathway susceptibly regulated by BFA. Together, these results support that GABARAP functions to stabilize GABA$_A$Rs by somehow promoting trafficking of GABA$_A$Rs from the Golgi to cell membranes, rather than by blocking endocytosis.

**Differential phosphorylation modulates the capacity of GABA$_A$ receptor γ2 subunit to interact with GABARAP vs. the clathrin adaptor protein AP2.** Guided by previous reports that

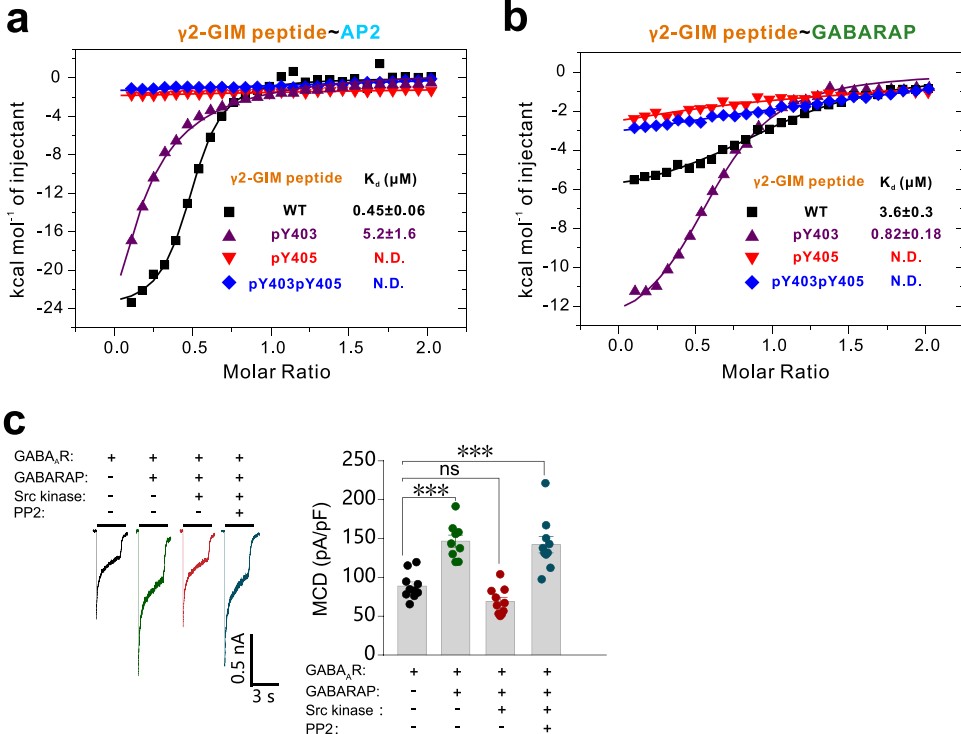

**Fig. 6 Differential phosphorylation modulates the capacity of GABA$_A$ receptor γ2 subunit to interact with GABARAP vs. the clathrin adaptor protein AP2. a** ITC-derived binding curves comparing binding affinities between AP2 and a synthesized γ2-GIM WT peptide or 3 phopho-peptides (pY403, pY405, and pY403pY405). **b** ITC-derived binding curves comparing binding affinities between GABARAP and the synthesized γ2-GIM WT peptide or 3 phopho-peptides (pY403, pY405, and pY403pY405). **c** Effects of Src kinase on GABA$_A$R-mediated current densities in HEK-293 cells co-expressing GABARAP. Representative trace records and average values of GABA currents activated by 1 mM GABA in HEK-293 cells co-expressing GABA$_A$R (α1β2γ2), GABARAP, and Src kinase with or without treatment of kinase inhibitor PP2. $n = 10, 9, 10$, and 10 of each group (from left to right). Data are represented as the mean ± SEM. ***$P = 0.0001$, GABA$_A$R vs. GABA$_A$R+GABARAP; ***$P = 0.0001$, GABA$_A$R vs. GABA$_A$R+GABARAP+Src kinase+PP2 based on one-way ANOVA; ns not significant ($P > 0.05$).

phosphorylation of the Y403 and Y405 residues can promote GABA$_A$R function (speculated to result from decreased AP2 binding)[34,35], we conducted ITC assays with synthetic γ2-GIM peptides (commercially synthesized 18-aa of the γ2-GIM including phosphorylated Y403, phosphorylated Y405, and double phosphorylated Y403 and Y405). The interaction between AP2 the synthetic pY403 γ2-GIM was dramatically weakened compared to binding with WT γ2-GIM ($K_d$ ~5.2 vs. ~0.45 μM) (Fig. 6a). In sharp contrast, the synthetic pY403 γ2-GIM showed a much stronger binding affinity for GABARAP compared to WT γ2-GIM ($K_d$ ~0.82 vs. ~3.6 μM) (Fig. 6b). Thus, the phosphorylation of Y403 on γ2-GIM dramatically alters both the binding affinity and binding-partner selectivity of γ2-GABA$_A$R. Both the pY405 and pY403pY405 peptides lost the abilities to bind either AP2 or GABARAP (Fig. 6a, b), indicating phosphorylation at Y405 residue is enough to abolish GABA$_A$R γ2 subunit in binding to AP2 or GABARAP, although the underlying in vivo regulation mechanism is largely unknown. These results with synthetic peptides were consistent with previous assays using phosphomimetic mutant variants including Y403E and Y405E (Supplementary Table 2).

To gain functional insights about the phosphorylation regulation, we performed kinase supplementation experiments by co-expression with the Src kinase, which was reported to phosphorylate the two tyrosine residues of γ2-GIM[36,37], and with or without further treatment by Src kinase inhibitor PP2[38] in the electrophysiological assays. Our results showed that co-expression with Src kinase blocked any GABARAP-mediated increase in GABA currents as induced by 1 mM GABA (Fig. 6c).

However, further treatment with PP2 inhibitor restored the increased effect of GABARAP on GABA currents (Fig. 6c), indicating PP2 treatment blocked Src kinase activity sufficiently. These findings clearly suggest the potential for the GABA$_A$R γ2 subunit in cells that can be controlled based on phosphorylation-mediated modulation of its binding affinity and selectivity.

**In vivo disruption of GABARAP–GABA$_A$R complex formation with inhibitory peptides from giant ankyrins deleteriously impacts GABAergic synaptic activity.** We next evaluated potential impacts from disrupting GABARAP–GABA$_A$Rs complex formation on GABAergic synaptic activity. Guided by our previous work, we conducted experiments using these potent Atg8-targeting inhibitory peptides from giant ankyrins[39]. The specific aim with these peptides was to induce interactions which efficiently prevent the GABARAP and GABARAPL1 proteins from binding with GABA$_A$R, as ankyrins bind to the overlapping region on GABARAP with the GIM interaction sites. We initially confirmed that the AnkG-WT and AnkB-WT peptides bind to GABARAP with a ~1000-fold and ~10,000-fold stronger affinity for GABARAP than γ2-GIM, respectively (Fig. 7a, b). The more GABARAPs-selective (compared to LC3s) AnkG-ER peptide also exhibit a ~1000-fold stronger affinity for GABARAP than γ2-GIM (Fig. 7a). In contrast, the AnkB-WR hardly binds with GABARAP (Fig. 7a), so AnkB-WR can be used as a specific negative control for comparisons against AnkB-WT and AnkG-ER in electrophysiological assays. We also confirmed that, in the

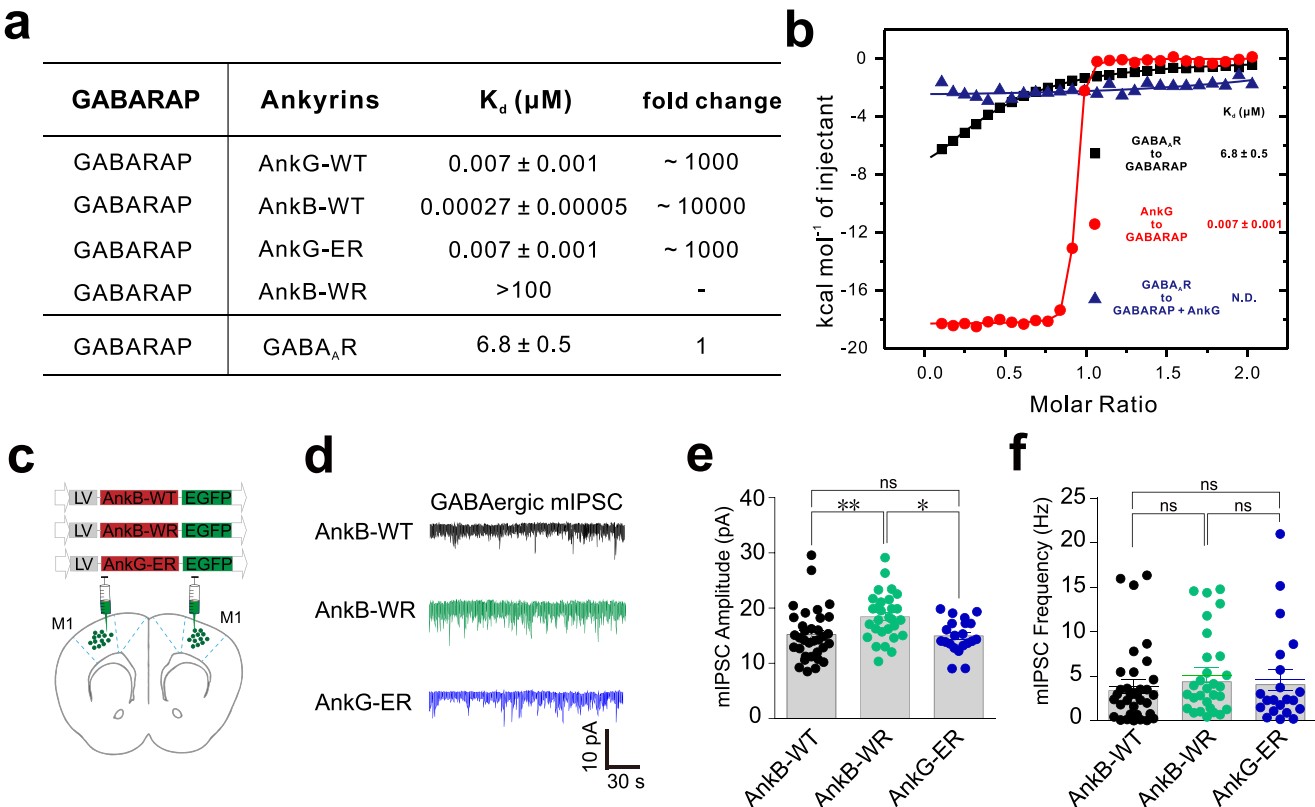

**Fig. 7 In vivo disruption of GABARAP–GABA$_A$R complex formation in M1 pyramidal neurons deleteriously impacts GABAergic synaptic activity. a** The measured binding affinities between various giant ankyrin peptides and GABARAP based on ITC and comparison against γ2-GIM in binding with GABARAP. **b** ITC-derived binding curves comparing the binding affinities between γ2-GIM and GABARAP or GABARAP–AnkG WT peptide complex. **c** Schematic diagram for in vivo experiments, showing the site of lentivirus injection for peptide expression and patch clamp recording. **d**–**f** Representative trace records **d**, average amplitude **e**, and average frequency **f** of GABAergic mIPSCs in primary motor cortex slices from male C56/BL mice. Data are represented as the mean ± SEM. $n = 34$, 29, and 21 of each group (from left to right) in **e** and $n = 34$, 29, and 21 of each group (from left to right) in **f**. $**P = 0.0079$, AnkB-WT vs. AnkB-WR; $*P = 0.0124$, AnkB-WR vs. AnkG-ER based on one-way ANOVA; ns not significant ($P > 0.05$).

presence of AnkG-WT, γ2-GABA$_A$R cannot interact with GABARAP (Fig. 7b).

We further showed that AnkB-WT peptide could also disrupt the GABARAP/γ2-GABA$_A$R interaction via GST pull-down experiment (Supplementary Fig. 7a). Moreover, we also examined that whether the γ2-GIM peptide could block GABARAP–GABA$_A$R complex formation in vivo, although the binding affinity between GABARAP and GABA$_A$R is modest. Our competition experiments showed that overexpression of γ2-GIM could indeed eliminate the GABARAP-mediated increase in GABA currents in HEK-293 cells (Supplementary Fig. 7b). Consistent with our biochemical data, AnkB-WT peptide also blocked GABARAP-mediated increase in GABA currents (Supplementary Fig. 7b), while AnkB-WT peptide itself had no effect on the GABA currents in the absence of GABARAP (Supplementary Fig. 7c). In consistent with these results, our cell surface biotinylation analysis of GABA$_A$R using AnkB peptide or γ2-GIM peptide showed that AnkB or γ2-GIM alone cannot promote increase of surface amount of GABA$_A$R, while γ2-GIM can block the GABARAP-mediated increase of GABA$_A$R membrane surface localization (Supplementary Fig. 7d).

Next, we chose giant ankyrin peptides as tools to evaluate effects from disrupting GABARAP–GABA$_A$Rs complex formation through brain slices recording assays. For these electrophysiological experiments, the GFP-AnkB-WT, GFP-AnkG-ER, and GFP-AnkB-WR (negative control) peptides were expressed individually in a mouse brain region, the primary motor cortex area (M1) as an example, by injection of corresponding lentivirus-based vectors (Fig. 7c). The mIPSCs from M1 pyramidal neurons in brain slices expressing GFP-AnkB-WT, GFP-AnkB-WR, or GFP-AnkG-ER were recorded using whole-cell patch-clamp techniques. Upon ankyrin-peptide-mediated (AnkB-WT and AnkG-ER peptides) disruption of GABARAP–GABA$_A$R complex formation, the mIPSC amplitudes were significantly reduced, suggesting a postsynaptic mechanism such as downregulation of membrane functional GABA$_A$Rs. In contrast, no reductions were observed upon lentivirus-mediated M1 pyramidal neuron expression of the negative control (AnkB-WR) peptide that does not affect GABARAP–GABA$_A$R complex formation (Fig. 7d, e). Note that the frequencies of mIPSCs in neurons expressing GFP-AnkB-WT and GFP-AnkG-ER peptides did not differ from those expressing GFP-AnkB-WR (Fig. 7d, f), suggesting no involvement of presynaptic elements. Collectively, these results indicate that disruption of GABARAP–GABA$_A$R complex formation in neurons specifically weakens postsynaptic GABA$_A$R currents.

## Discussion

In this study, we revealed the interactions underlying binding between γ2-GABA$_A$R and GABARAP. We also confirmed the function of GABARAP in trafficking GABA$_A$R to the cell surface. Although there is overlap in the γ2-GABA$_A$R ICD sequences which, respectively, bind with GABARAP and the clathrin adaptor protein AP2, our solved crystal structure enabled identification of the residues governing the specific interactions of γ2-

GABA$_A$R with these two proteins. Interestingly, we discovered that differential phosphorylation of tyrosine residues of γ2-GIM determine its capacity to interact with GABARAP vs. AP2. Finally, we demonstrate in vivo that the amplitudes of mIPSCs for GABAergic synapses are reduced by blocking GABARAP–GABA$_A$R complex formation.

As a member of the Atg8 family, many studies have explored potential autophagic roles for GABARAP. However, it bears emphasis that GABARAP was initially described as a binding partner of GABA$_A$R, and several related questions about the apparent non-autophagic roles of GABARAP remain unanswered. It has been speculated that functional redundancy between GABARAP and GABARAPL1 has complicated experimental investigations seeking the exact roles of these two proteins in the nervous system. Our work in the present study demonstrates the GABARAP and GABARAPL1, in addition to their known autophagic roles, differ from other Atg8 family members in their binding with GABA$_A$R and functioning in the delivery pathway for receptors to the surface of neurons. However, we cannot exclude the possibility that GABARAP and/or LC3s may also exert vital autophagy-related functions in neurons, as all the family members are found in the axons of mammalian neurons[40,41]. Considering the extremely long lifespan of myelinated axons in animals, it is possible that neurons may employ as-yet-unknown autophagy-related processes which are simply not needed in other cell types. As GABARAP and GABARAPL1 are already known to participate in multiple steps of autophagy, perhaps their further study in neurons may help to uncover previously unknown, axon-specific cellular recycling processes.

The AP2 complex functions on the cell membrane to internalize receptors in clathrin-mediated endocytosis, which is critical for the regulation of cell surface level of GABA$_A$Rs. Several studies have already showed that AP2 can also bind to β subunits of GABA$_A$R in a phosphor-dependent manner[42,43]. In our study, we showed that the AP2-binding motif on γ2-GABA$_A$R overlaps with GABARAP-binding region. Interestingly, differential phosphorylation at distinct tyrosine residues can switch the binding abilities of GABA$_A$R γ2 subunit to GABARAP and AP2, respectively, giving effective modulation mechanism of PTM. Given that there are sufficient amount of kinases in the postsynaptic densities of inhibitory synapses, this phosphorylation-mediated switching of binding with distinct partners for GABA$_A$R is an effective regulatory mechanism.

GABAergic synapses are the only type of synapses present in the axon initial segment (AIS) of neurons. The AIS is characterized by enrichment of voltage-gated ion channels, adhesion molecules, scaffold proteins, and regulatory proteins including diverse kinases. The Ankyrin-G scaffold is known to be a major organizer of the AIS protein complex. Previous in vivo studies have revealed that giant ankyrin-G functions to stabilize the localization of GABA$_A$R at membrane sectors of both the AIS and the somatodendritic regions of neurons[44,45]. Our demonstration of ankyrin peptides as powerful tools for empirically characterizing GABARAP–GABA$_A$R complex functions implies that further engineering of giant ankyrins or chimeric variants could help to more precisely define the trafficking networks at and beneath the cell membrane micro-domains of neurons. GABAergic synaptic activity is inarguably one of the most impactful forms of neurotransmission underlying animal behavior, and our present work clearly reviews the interaction between GABA$_A$R and its binding partner GABARAP and their critical functions in this fundamental process.

## Methods

### Constructs, protein expression, and purification.
The coding sequences of the GABARAP (UniProt: Q9DCD6), GABARAPL1 (UniProt: Q8R3R8), GABARAPL2 (UniProt: P60521), LC3A (UniProt: Q91VR7), LC3B (UniProt: Q9CQV6), and LC3C (UniProt: Q9BXW4) constructs were PCR amplified from mouse muscle or brain cDNA libraries[39]. The full-length α1 subunit (UniProt: P62812) and full-length γ2 subunit (UniProt: P22723-2) of GABA$_A$R genes were PCR amplified from a mouse cDNA library (Supplementary Table 3). The full-length β2 subunit of GABA$_A$R (UniProt: P47870) gene was PCR amplified from a human cDNA library (Supplementary Table 3). The full-length AP2 μ1 subunit (UniProt: P84092) gene was PCR amplified from a rat cDNA library (Supplementary Table 3). Various mutations or shorter fragments of α1 subunit, β2 subunit, and γ2 subunit of GABA$_A$R, GABARAP, GABARAPL1, and AP2 were generated using standard PCR-based methods and confirmed by DNA sequencing. All of these coding sequences were cloned into a home-modified pET32a vector for protein expression. The N-terminal thioredoxin-His$_6$-tagged proteins, N-terminal MBP-His$_6$-tagged proteins, and N-terminal His$_6$-tagged proteins were expressed in Escherichia coli BL21 (DE3) cells in LB medium at 16 °C and purified using a nickel-NTA agarose column followed by size exclusion chromatography (Superdex 200 or Superdex 75) with a column buffer containing 50 mM Tris, 100 mM NaCl, 1 mM EDTA, and 1 mM DTT at pH 7.8. The thioredoxin-His$_6$ tag was removed by incubation with HRV 3C protease and separated by size exclusion columns when needed. All the phosphor-γ2-GIM peptides were commercially synthesized (GL Biochem (Shanghai) Ltd).

### ITC assay.
ITC measurements were carried out on a VP-ITC Microcal calorimeter (Malvern) at 25 °C. All proteins were in 50 mM Tris buffer containing 100 mM NaCl, 1 mM EDTA, and 1 mM DTT at pH 7.8. Each titration point was performed by injecting a 10 μL aliquot of γ2 subunit protein or peptide (100–200 μM) into GABARAP, GABARAPL1, or AP2 protein samples (10–20 μM) in the cell at a time interval of 180 s to ensure that the titration peak returned to the baseline. The titration data were analyzed using the program Origin7.0 and fitted by the one-site binding model.

### Analytical gel filtration chromatography.
Analytical gel filtration chromatography was carried out on an AKTA FPLC system (GE Healthcare). Recombinant proteins with N-terminal His$_6$-tag were concentrated to 100 μM and loaded onto a Superose12 or Superdex200 increase 10/300 GL column (GE Healthcare) equilibrated with the assay buffer (50 mM Tris, pH 7.5, 100 mM NaCl, 1 mM EDTA, and 1 mM DTT). The data were analyzed using the Origin 7.0.

### GST pull-down assay.
For GST pull-down assays, 2 μM GST-tagged γ2 subunit 398–415 or equimolar GST-tagged γ2 subunit 398-415 Y403E (or GST as the negative control) was first incubated with the purified proteins of GABARAP and AP2 for 1 h at 4 °C. The 30 μl GSH-Sepharose 4B slurry beads in an assay buffer (50 mM Tris, pH 7.5, 100 mM NaCl, 1 mM EDTA, and 1 mM DTT) were then incubated with the mixture for 30 min at 4 °C. After three times washing, the captured proteins were eluted by boiling, resolved by 10% SDS–PAGE, and detected by Coomassie blue staining.

GFP-tagged WT and mutants of γ2 subunit were overexpressed in HEK-293 cells. Cells were harvested and lysed by the ice cold cell lysis buffer (50 mM Tris pH 7.4, 150 mM NaCl, 1% NP-40, and protease inhibitor cocktail) supplemented with 1% (w/v) n-dodecyl-β-D-maltopyranoside (DDM, Anatrace). After centrifugation at 16,873 × g for 10 min at 4 °C, the supernatants were incubated with 2 μM of various WT or mutants of GST-GABARAP, GST-GABARAPL1, or GST-LC3A for 1 h at 4 °C. The 30 μl GSH-Sepharose 4B slurry beads in lysis buffer were then incubated with the mixture for 30 min at 4 °C. After three times wash with the cell lysis buffer, the captured proteins were eluted by 20 μl SDS–PAGE loading dye and detected by immunoblotting.

### Surface biotinylation.
Transfected HEK-293 cells were biotinylated with 0.25 mg/ml Sulfo-NHS-LC-biotin (Thermo Scientific) in phosphate-buffered saline containing Mg$^{2+}$/Ca$^{2+}$ for 45 min at 4 °C followed by quenching with glycine for 20 min. The cells were lysed in RIPA buffer (50 mM Tris pH 7.5, 150 mM NaCl, 1% Triton X-100, 1 mM PMSF, 1 mg/ml aprotinin, leupeptin, and pepstatin), and the lysates were incubated with NeutrAvidin beads (Thermo Scientific) overnight at 4 °C. After three times wash with the RIPA buffer, the captured proteins were eluted by 20 μl SDS–PAGE loading dye and detected by immunoblotting.

### Immunoblotting.
The cell lysates or beads were incubated in SDS–PAGE loading buffer for 20–30 min at room temperature. The samples were separated on SDS–PAGE, transferred, probed with GFP (1:1000; Proteintech 66002-1-Ig), Flag (1:2000; Proteintech 66008-3-Ig), or GAPDH (1:2000; Proteintech 10494-1-AP) antibodies and visualized with enhanced chemiluminescence (Amersham Biosciences). The intensity of immunoreactive bands was analyzed using the Image-Pro Plus 5.1 software (Media Cybernetics, Inc.). All experiments were repeated at least three times.

### Crystallography.
Crystallization of the γ2 398-415/GABARAPL1 complex was performed using the hanging drop vapor diffusion method at 16 °C. Crystals of the GABARAPL1–γ2-GIM were obtained from the crystallization buffer containing

0.2 M ammonium acetate, 0.1 M sodium citrate tribasic dehydrate pH 5.6, and 30% PEG 4000. The diffraction data were collected at Shanghai Synchrotron Radiation Facility and processed and scaled using HKL2000[46].

Structures were solved by molecular replacement using PHASER[47] with the apo-form structures of GABARAPL1 (PDB: 2R2Q) as the searching models. Peptides were manually built according to the $F_o − F_c$ difference maps in COOT[48]. Further manual model adjustment and refinement were completed iteratively using COOT[48] and PHENIX[49] or Refmac5[50]. The final model was validated by MolProbity[51]. The final refinement statistics are summarized in Supplementary Table 1. All structure figures were prepared by PyMOL (http://www.pymol.org). The coordinates and structure factors of the GABARAP and γ2-GABA$_A$R complex have been deposited in the Protein Data Bank with accession number 7CDB.

**Animals**. C57BL/6 male mice (7–8 weeks old) were used and obtained from Vital River Laboratory Animal Technology Co., Ltd. (Beijing, China). All mice were housed under a normal 12-h dark/light cycle at 22–24 °C and (40–70%) RH and with free access to water and food. All procedures were approved by the Institutional Animal Use and Care Committee of School of Life Sciences, University of Science and Technology of China.

**HEK-293 cells culture**. HEK-293 cells were cultured as described previously[52]. In brief, the cells were seeded at a density of $10^6$ cells/ml in 35 mm disposable corning dishes with a medium composed of DMEM, 10% fetal bovine serum, and penicillin–streptomycin 100 U/mL in an incubator at 37 °C in a humidified atmosphere containing 5% $CO_2$. Cells were then allowed to grow to 70% confluence before subculturing using Trypsin at 0.25% (w/v). The medium was changed every 2 days.

**Electrophysiological recording**. Plasmids coding WT or various mutant GABA$_A$R and GABARAP were co-transfected into HEK-293 cells with Lipofectamine 2000 (Invitrogen) reagents according to the manufacturer's instructions. Electrophysiological recordings were performed 48 h after transfection. HEK-293 cells were firstly treated with 0.25% (w/v) trypsin 2 h before patch-clamp recording. The recording was carried out with an external solution containing 140 mM NaCl, 5 mM KCl, 2.0 mM CaCl$_2$, 1.0 mM MgCl$_2$, 10 mM glucose, and 10 mM HEPES (pH 7.4 with NaOH, ~320 mOsm with sucrose). The recording pipettes were filled with intracellular solution containing 140 mM CsCl, 4 mM MgCl$_2$, 10 mM EGTA, 10 mM HEPES, 0.5 mM Na-GTP, and 2 mM Mg-ATP (pH 7.2 with CsOH, ~280 mOsm). The input resistances amounted to 3–5 MΩ. GABA at 1 mM was used to induce membrane currents. The membrane currents were recorded in the whole-cell configuration using an Axopatch 200B amplifier (Axon) at 20–25 °C. Cells were always held at –60 mV. Series resistance compensation of 60–80% was used and monitored in the whole-cell recording experiments. Data were acquired and analyzed using pClamp 10.4 software (Molecular Devices, Sunnyvale, CA). MCD (p$A$/p$F$) was determined by dividing the GABA evoked peak current by the cell membrane capacitance. The Warner fast-step stepper motor-driven system was used to apply GABA. Dynasore at 80 μM was applied in the medium 2 h before recording as needed. The Src family tyrosine kinase inhibitor PP2 at 10 μM was added in the medium 30 min before recording. Brefeldin A at 5 μg/mL was added in the medium 30 min before recording to specifically block ER–Golgi protein trafficking.

**Stereotactic surgery and virus injection**. To achieve specific targeting of M1, a previously described protocol was used[53]. Briefly, C57BL/6 male mice (7 weeks old) were anesthetized with 1–2% isoflurane and placed in a stereotaxic apparatus following with a stable depth of anesthesia. A scalp incision and craniotomy were made with a dental drill. A volume of 100 nL virus (Shanghai Genechem Co., Ltd.) expressing AnkB-WT (VEEEWVIVSDEEIEEARQKAPLEITEY), AnkB-WR (VEEE RVIVSDEEIEEARQKAPLEITEY), or AnkG-ER (PEDDWTRFSSEEIR-EARQAAAS HAPS) peptides was, respectively, injected to the M1 region. The following stereotaxic coordinates were used for M1: AP: 1.20; ML: ±1.8 mm; DV: 0.60 mm from dural surface. The micropipette was made to stay in place for 5 min before being slowly retracted after injection. Following injections, the scalp incision was then closed using tissue glue and mice were returned to their home cages, and allowed to recover for 3 weeks before slice recordings.

**Brain M1 slice preparation and recording**. For primary motor cortex (M1), slices from C57BL/6 male mice (10 weeks old) were prepared as followings: coronal section of M1 slices (280-μm thick) were prepared with Leica Vibratome using ice-cold cutting solution containing 30 mM NaCl, 26 mM NaHCO$_3$, 10 mM glucose, 194 mM sucrose, 4.5 mM KCl, 1.2 mM NaH$_2$PO$_4$, 1 mM MgCl$_2$ and continuously bubbled with carbogen (95% O$_2$−5% CO$_2$). After section, slices were then transferred to a perfusion chamber containing artificial cerebrospinal fluid (ACSF): 124 mM NaCl, 4.5 mM KCl, 1 mM MgCl$_2$, 2 mM CaCl$_2$, 1.2 mM NaH$_2$PO$_4$, and 26 mM NaHCO$_3$, which bubbled in carbogen continuously. Slices were transferred to recording chamber after 60 min recovery at room temperature. The chamber was continuously perfused with ACSF (2–3 ml/min). Recordings were carried out at 34 °C using glass pipettes filled with internal solution containing 120 mM CsCl,

4 mM MgCl$_2$, 10 mM EGTA, 10 mM HEPES, 0.5 mM Na-GTP, and 2 mM Mg-ATP (pH 7.2 with CsOH, ~280 mOsm). For mIPSCs recording, 4 mM kynurenic acid, 1 μM strychnine, and 10 μM TTX were added into continuously perfused ACSF solution. The input resistance (3–5 MΩ) was monitored continuously. Data were acquired and analyzed using pClamp 10.4 software (Molecular Devices, Sunnyvale, CA).

## Data availability
The data supporting the findings of this manuscript are available from the corresponding author upon reasonable request. The coordinates and structure factors of the GABARAPL1 and γ2-GABA$_A$R complex have been deposited in the Protein Data Bank with accession number 7CDB [https://doi.org/10.2210/pdb7CDB/pdb]. Source data are provided with this paper.

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

## Acknowledgements

We thank the Shanghai Synchrotron Radiation Facility (SSRF) BL17U1, BL18U1 and BL19U1 for X-ray beam time. This work was supported by grants from the Ministry of Science and Technology of the People's Republic of China (2019YFA0508402, to C.W. and M.Z.; 2016YFC1300500-2, to W.X.), the National Natural Science Foundation of China (31670734 and 91953110, to C.W.; 21907033 and 31971140, to J.L.; 91849206 and 91649121, to W.X.; 81901157, to G.Z.), the Innovative Program of the Development Foundation of Hefei Center for Physical Science and Technology (2018CXFX008, to C.W.; 2017FXZY006, to W.X.), the Guangdong Province Key Project (2018B030335001, to M.Z.), the Strategic Priority Research Program of the Chinese Academy of Sciences (XDB39050000, to W.X.), China Postdoctoral Science Foundation (2020TQ0314, to G.Z.), the CAS Key Research Program of Frontier Science (ZDBS-LY-SM002, to W.X.), the CAS Interdisciplinary Innovation Team (JCTD-2018-20, to W.X.), and the Fundamental Research Funds for the Central Universities. C.W. is supported by the Chinese Academy of Sciences Pioneer Hundred Talents Program.

## Author contributions

C.W., J.L., and M.Z. conceived research; C.W., J.L., J.Y., W.X., and M.Z. designed research; J.Y., R.Z., C.K., J.L., and C.W. performed biochemical, structural, and chemical biology experiments; G.Z. and C.M. performed electrophysiological experiments; all authors analyzed data; J.Y. and C.W. wrote the paper with inputs from G.Z., J.L., R.Z., W.X., and M.Z., and all authors approved the manuscript; C.W. coordinated the project.

## Competing interests

The authors declare no competing interests.

## Additional information

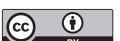

