## [Peer Review File · Nature Communications]

REVIEWER COMMENTS

Reviewer #1 (Remarks to the Author):

GABA-A-receptors (GABA-A-Rs) are the principal mediators of rapid inhibitory synaptic transmission in the central nervous system. The proper level of GABA-A-Rs on the neuronal plasma membrane is important for their functions, which is regulated by two main mechanisms, one is the transport of GABA-A-Rs from Golgi to the plasma membrane and another is endocytosis-mediated recycling and degradation of GABA-A-Rs. GABARAP is a ubiquitin-like protein that was identified as a binding protein for GABA-A-Rs and is considered to promote the transport of GABA-A-Rs to the plasma membrane although the molecular mechanism remains to be elucidated.

Here the authors identified the GABARAP-binding sequence (LIR) in gamma2-GABA-A-R, which is different from the previously reported binding region, and determined the crystal structure of the LIR complexed with GABARAP-L1, a homolog of GABARAP, showing that GABARAP competes with AP2, a clathrin adaptor protein, for GABA-A-R binding. Then the authors performed ITC using mutant proteins and confirmed that the observed hydrophobic interaction in the crystal is important for the affinity. The authors then studied the biological significance of the GABARAP-LIR interaction by whole-cell patch clamp experiments and proposed that the interaction is important for increasing the membrane level of GABA-A-Rs via a trafficking pathway. Finally, the authors demonstrated that the amplitudes of miniature inhibitory postsynaptic currents for GABAergic synapses are reduced by treatment of the cells with peptides that inhibit the GABARAP-LIR interaction. The structural and interaction analyses have been well performed and clearly established the structural basis of the GABARAP-family interaction with gamma-GABA-A-R. Biological assays seem to support the importance of the interaction in GABA-A-Rs functions. These data will contribute to unveil yet-mysterious functions of GABARAP-family in synaptic transmission. However, there are some critical concerns in the functional analyses, which must be resolved prior to be published in Nature Communications.

Major points

- 1) In the crystal structure, electrostatic interaction was observed between GABARAP K24 and LIR E402 and between GABARAP R67 and LIR E406. In addition, hydrogen bond was observed between GABARAP E17 and LIR Y405 (main-chain) and between GABARAP K66 and LIR D412 (main-chain). Nevertheless, the authors did not perform mutational analyses on them and instead performed mutational analysis at GABARAP R28, which was not involved in the interaction in the crystal structure. Perform mutational analyses on these residues using ITC and study whether the hydrophilic interactions observed in the crystal are important for the affinity.
- 2) The authors concluded that GABAA-R-gamma2 LIR is selective toward GABARAP and GABARAP-L1 among 6 homologs. Until now many structures have been reported for mammalian Atg8s complexed with various LIRs, which unveiled preference of LIRs for GABARAP and LC3 family. For example, GABARAP interaction motif (GIM) was proposed as a preferred sequence for GABARAP family (PMID 30518659). Compare the determined structure with previously reported structures and discuss the mechanism of how GABAA-R-gamma2 LIR selectivity binds GABARAP and GABARAP-L1 among six Atg8 homologs.
- 3) By the observation that the treatment of endocytosis inhibitor dynasore caused similar increase of GABA MCD for cells with and without expression of both GABA-A-R and GABARAP, the authors concluded that binding of GABARAP to GABA-A-R promotes the trafficking of the receptor from Golgi to cell membrane rather than inhibiting endocytosis. However, in this study the authors did not confirm that dynasore treatment alone blocked endocytosis sufficiently, and so there remains a possibility that endocytosis was partially inhibited by the expression of GABA-A-R and GABARAP, which was further inhibited by dynasore treatment. Perform back-up experiments, for example confirming the sufficient endocytosis inhibition by dynasore treatment or performing similar experiments using the cells AP2 is knocked-down.

Minor points

- 1) In Fig. 1b, legend, "gamma2-ICD is highlighted in red", red is not used in the figure.
- 2) Provide raw data and N (stoichiometry of binding) values for all ITC data (Fig. 1c, d, 2d, e, 3b, d, e, 5a, b, Supple Table2).

Reviewer #2 (Remarks to the Author):

This study outlines some new perspectives on the association of GABARAP with the GABAA receptor and identifies a new binding motif, which overlaps with the AP2 protein responsible for receptor internalisation. It is proposed that the competition between these proteins, which interact with the intracellular loop of gamma2 subunits, dictates the stability of the membrane bound receptor, and provides a plasticity 'handle'. Besides binding affinity data, crystallographic evidence is presented for the GABARAP interaction, as well as functional experiments that are used to establish the cellular role of GABARAP in the GABAA receptor's ability to access and remain in the cell membrane. This is a good but preliminary study. It re-addresses potential oversights in the association between two key proteins which affect neuronal activity, but there are a significant number of major issues which detract from the study's quality.

1. The structural evidence for a novel binding site between GABARAP and the gamma2 loop (G2-LIR) seems plausible, though the three key observations used from the literature to question current opinion on the role of GABARAP are selective and arguably tenuous. That said, they do provide a reason for this study. However, the general conclusions pertaining to GABARAP's neuronal function do not differ much from those already published. The more novel aspect of the paper, that exploring the interaction and competition between GABARAP and AP2, and the potential to tune receptor levels in the membrane, potentially via phosphorylation, and affect plasticity, is insufficiently explored. I have highlighted many questionable areas of the study below.

2. Interaction site/structure

Crystal structures for GABARAP, AP2 and the gamma2 LIR (in various combinations) already exist. This detracts from the impact of the crystallographic data, although breakdown of the key residues which participate in the GABARAP:G2-LIR interaction (and their subsequent mutational evidence) is interesting and novel. The previously reported G2-LIR:AP2 complex crystal structure (which is incorrectly cited in the text) has much overlap with this study in reporting a similar interaction site.

The Atg8 family of proteins (of which GABARAP is a member) have 'high sequence similarity' yet only two bind to the G2-LIR. It would be useful to have more information as to why this is. Notably, the L1 variant of GABARAP binds equally well, but reasoning as to why the native (and more physiologically relevant) GABARAP was not crystallized is absent. We need to be better informed about the degree of sequence similarity between these variants, as there is total reliance on L1 for all of the structural interpretation, not only with LIR, but also with AP2 binding. Given this reliance, it needs to be established if GABARAP and GABARAPL1 have similar effects in the key functional e-phys experiments. If these controls are not present, then the validation of the G2-LIR:GABARAPL1 X-stal model is questionable.

Also, whilst performing the crystallizations, it would have been prescient to co-crystallize the previously reported 424-441 peptide, to verify non-binding in this different format.

3. Pull-downs and binding

There is an over-reliance on the ITC technique (and analytical gel filtration), which is used throughout

the study, for establishing the binding interaction, and it may present interpretational challenges. The ITC technique is sensitive, but the LIR peptide is derived from a membrane-bound protein so its structure in solution may be artificial. The authors will know that many previous studies have very successfully used GST pull-down assays to study this interaction. This latter technique should be more prominent here with regard to quantifying binding of the novel protein motif/peptide, as the initial starting structures for the peptide(s) from these techniques may vary more than anticipated. It will satisfy readers who feel that abject dismissal of the previously published peptide as a binding motif is premature. It would be useful to have ITC calorimetry data for the full length isolated protein (membrane-bound?) for comparison to the isolated peptide.

4. G2-LIR mutations

Please give some reasoning behind the selected mutations for this peptide, as this is unclear. Very few seem to conserve charge (eg R28E), whilst others introduce charge (which may explain the increase in binding with Y403E). The appropriateness of the mutations needs rationalising - were just key (interacting?) residues mutated, or was this extended to other/all residues in the peptide. Also, did the K_d remain unchanged for supposedly unimportant residue mutation (see below)?

Given that the G2-LIR peptide is only 18aa long, some mutations may affect the overall backbone of the peptide, and could be the reason why some mutations are more/less effective than others. Some comment about this is needed.

5. Main functional data.

The conclusions from this data largely echo those currently in the literature (whilst noting the novelty of the binding of the LIR(398-415) peptide), ie the role of GABARAP is unchanged. Much of the data to get to this point is still quite preliminary. The co-expression studies need to have more controls - i.e. concentration-response curves to establish that there are no changes in GABA potency and/or GABA efficacy in the presence of the peptides. There needs to be at least some basic level of immunocytochemistry in place to verify increases in receptor number in both HEK cells and neurons, and checks with Golgi markers to quantitate if trafficking is facilitated (or not) in the presence of GABARAP and the peptides.

Cell recordings are made using a maximal dose of GABA on cells which likely express $\alpha 1\beta 2\gamma 2$ receptors very efficiently. If these cells are not compensated for series resistance (not reported), they will be prone to underestimation of the maximal currents, which is the only metric used in these experiments. Not every GABA-A receptor in the neuron is the same, so co-expression with an extrasynaptic receptor type should be included for a few of the conditions.

6. Phosphorylation

It is not clear how phosphorylated forms of Y403 & 5 are generated in the study. The result (line 272) that phosphorylated Y405 and Y403/5 means neither AP2 nor GABARAP bind the G2-LIR is puzzling. Given that these two residues are both tyrosines so are likely to be equally susceptible to phosphorylation/de-phosphorylation by kinases/phosphatases, it is hard to reconcile how only phosphorylation of Y403 (not Y405) is likely to happen in neurons to permit GABARAP/AP2 binding. These phosphorylation experiments (not simply phospho-dead/mimetic mutations, but also kinase supplementation experiments) need to be undertaken in the functional ephys assays to understand the complete effects of these changes, and to give them a more relevant context.

7. Ankyrins

The competition between GABARAP & AP2 for binding sites, and their functional consequences, are important to understand for this study. However, this interesting concept is completely overlooked. Instead the authors opt to disrupt the G2-LIR:GABARAP interaction site with ankyrins. These attachment proteins were chosen because the modest binding of G2-LIR peptide (as a competitive inhibitor) was deemed to be a 'sub-optimal tool'. This competition experiment should have at least

have been tried (even in HEK cells). The ankyrins chosen instead disrupt GABARAP binding much more effectively. However, we are not told where they bind, as, if it is not overlapping with the LIR site, then they are not wholly appropriate. Also, cortical neurons are used to test the effectiveness (and physiological consequences) of disrupting the GABAA-R:GABARAP interaction, yet this interaction needs controls to be performed in HEK cells (at least) before their consequences can be properly interpreted in neurons following viral transfections. Do ankyrins even affect the behaviour of GABAA-Rs in HEK cells in the absence of GABARAP? Generally, the experiments associated with figure 5 are poorly controlled.

8. Methods, n numbers, statistics, figures

Many of the figures/data are presented as if they are an $n=1$. Clearly more than one experiment has been performed in most cases as K_d values are \pm errors. The text, legends and methods have no (or very few) N numbers reported, and no statistical analysis, so we cannot assess how robust the results are.

Many of the supplementary data figures will be meaningless to many readers. The fluorescence assay (Sfig1) is difficult to interpret for quantitation. Similarly for sFig3 & 4 chromatography plots. Both these assays are accompanied by insubstantial Methods sections which lack the necessary detail expected of the journal, and do not help interpretation of these figures. The Methods section on the crystallography also falls far short on detail that is expected, and many other Methods sections are similarly lacking.

Figure 3a does not clearly represent what is being discussed in the text. This peptide complex structure (PDB code 2PR9) needs to be cited. I don't think this is a model generated by this study (it is not clear), so this citation needs to be present. If the model is derived here, then no docking methodology is included in the text. This data's origin is confusingly presented.

The PDB code for the new Xstal complex needs to appear in the Methods (as well as fig legend).

Fig 5f states that the treatments do not change frequency. This is the only figure where any statistics are presented yet the foundation for this is based on data which changes in frequency from <1 Hz to 20 Hz. The Methods do not say at what age slice recording was happening, so this variability (which needs scrutiny) may result from very different age animals.

Some labelling in sFig2 is not correct. The gel in sFig4b looks overloaded or over-exposed?

Reviewer #3 (Remarks to the Author):

The manuscript by Jin Ye et al. describes the structure of the complex between GABARAPL1 and a peptide corresponding to the GABARAP-interacting region of $\gamma 2$ -GABAAR. It also reports extensive complementary experiments supporting the structural findings and describing the details of the GABARAP/ $\gamma 2$ -GABAAR interactions in vivo and in vitro. The experiments are well designed and well conducted; the data is compelling, interesting, and advances the understanding of how inhibitory synaptic transmission is regulated. Unfortunately the results are significantly overinterpreted. The most glaring examples are outlined below:

- The authors claim in the Abstract, Results Section and Discussion that they "demonstrate that GABARAP functions to stabilize GABAARs via promoting its trafficking pathway instead of blocking receptor endocytosis". This is not really the case. While the data is more consistent with GABARAP promoting GABAR trafficking, than with blocking receptor endocytosis, it does not prove the former, just suggests it. The facts that: 1: the endocytosis inhibitor dynasore blocks some of the receptor endocytosis (authors say previous studies show a $\sim 90\%$ block, but do not show how much it actually

blocks in the assay they use), and 2: addition of GABARAP further increases the GABA MCDs, do not demonstrate that GABARAP does not block endocytosis. Indeed, the findings could also be explained by GABARAP blocking the remaining (say 10%) receptor endocytosis. The authors do not directly show that GABARAP increases receptor trafficking.

- The model for GABARAP function that the authors propose in the Discussion Section (line 401 – till the end) is solely based on: "According to our results, giant ankyrin-G cannot form a triple complex with GABARAP and GABAAR, but rather competes out GABAAR by binding to the same region of GABARAP with super-strong affinity". But the authors never actually show this in this manuscript. They only show that the ankyrin peptides reduce the mIPSC amplitudes and assume that this reduction is due to the peptides blocking the GABARAP/GABAAR interaction based on the previously published papers describing these peptides: line 329: "we conducted experiments using previously reported potent Atg8-targeting inhibitory peptides from giant ankyrins ... these peptides induce interactions which prevent the GABARAP and GABARAPL1 proteins from binding with GABAAR". So everything that the authors discuss and propose starting in line 401 – till the end, is solely based on previously reported experiments and not on the data reported in this manuscript.

I suggest the authors revise their manuscript so that their conclusions and interpretations match the presented data.

Minor points:

- The manuscript needs editing to improve the use of the English language (including use of commas, tenses, "the"/"a" etc.); for example what does "our results uncover the mechanism governing the binding site between GABAAR and GABARAP" (line 46) mean? What is a mechanism that governs a site?
- The authors have named their peptide γ 2-LIR for LC3-Interacting Region, but this peptide/region actually does not interact with any of the LC3s (line 174).

Point-by-point responses to the comments raised by the reviewers

Before point-by-point responses to the referees' comments, we thank all the reviewers for recognizing the novelty and interests of our works and their critical and constructive suggestions and guidance that help us to efficiently improve our manuscript. Our responses are shown in blue.

Reviewer #1 (Remarks to the Author):

GABA-A-receptors (GABA-A-Rs) are the principal mediators of rapid inhibitory synaptic transmission in the central nervous system. The proper level of GABA-A-Rs on the neuronal plasma membrane is important for their functions, which is regulated by two main mechanisms, one is the transport of GABA-A-Rs from Golgi to the plasma membrane and another is endocytosis-mediated recycling and degradation of GABA-A-Rs. GABARAP is a ubiquitin-like protein that was identified as a binding protein for GABA-A-Rs and is considered to promote the transport of GABA-A-Rs to the plasma membrane although the molecular mechanism remains to be elucidated.

Here the authors identified the GABARAP-binding sequence (LIR) in gamma2-GABA-A-R, which is different from the previously reported binding region, and determined the crystal structure of the LIR complexed with GABARAP-L1, a homolog of GABARAP, showing that GABARAP competes with AP2, a clathrin adaptor protein, for GABA-A-R binding. Then the authors performed ITC using mutant proteins and confirmed that the observed hydrophobic interaction in the crystal is important for the affinity. The authors then studied the biological significance of the GABARAP-LIR interaction by whole-cell patch clamp experiments and proposed that the interaction is important for increasing the membrane level of GABA-A-Rs via a trafficking pathway. Finally, the authors demonstrated that the amplitudes of miniature inhibitory postsynaptic currents for GABAergic synapses are reduced by treatment of the cells with peptides that inhibit the GABARAP-LIR interaction. The structural and interaction analyses have been well performed and clearly established the structural basis of the GABARAP-family interaction with gamma-GABA-A-R. Biological assays seem to support the importance of the interaction in GABA-A-Rs functions. These data will contribute to unveil yet-mysterious functions of GABARAP-family in synaptic transmission. However, there are some critical concerns in the functional analyses, which must be resolved prior to be published in Nature Communications.

We thank the reviewer for the careful reading of our study and for appreciating the value from our combination of biochemical, structural biology, and electrophysiological approaches to investigate fundamental questions about regulation of GABA_AR membrane localization through GABARAP. We are also grateful for the extremely helpful guidance about how to improve the quality and purport of our study with additional confirmatory experiments. We have now addressed all the reviewer's concerns for both the structural and functional analyses, and have revised our manuscript to incorporate our new data and the Reviewer-directed changes in emphasis.

Major points

1) In the crystal structure, electrostatic interaction was observed between GABARAP K24 and LIR E402 and between GABARAP R67 and LIR E406. In addition, hydrogen bond was observed

between GABARAP E17 and LIR Y405 (main-chain) and between GABARAP K66 and LIR D412 (main-chain). Nevertheless, the authors did not perform mutational analyses on them and instead performed mutational analysis at GABARAP R28, which was not involved in the interaction in the crystal structure. Perform mutational analyses on these residues using ITC and study whether the hydrophilic interactions observed in the crystal are important for the affinity.

We totally agree with the reviewer's suggestions to experimentally evaluate the hydrophilic interactions we observed in the crystal structure. In the revised manuscript, we have added assay data for the following variants: GABARAP K24E, R67E, E17A, and K66E (Figure I). Consistent with our structural observations, K24E variant profoundly reduced binding between GABARAP and LIR; R67E totally abolished the interaction; E17A slightly weakened the interaction, and K66E weakened the interaction by around 4 times. Collectively, our new ITC data for the GABARAP variants (K24E, R67E, E17A, and K66E) and our original data (Y49A and L50A) in binding with LIR validate our interpretation from the crystal structure and are consistent with our conclusions from the ITC data for the LIR variants (E402T, E406R, Y405E, and L408Q) in binding with GABARAP.

Figure I: ITC results showing that mutating the key residues in GABARAP decrease the binding to $\gamma 2$ -LIR when compared to the wild-type GABARAP. The error of K_d for each ITC curve represents the curve fitting error. N: stoichiometry of binding. The 'N.D.' denotes that there is no detectable binding.

In addition, we have now performed ITC assays with several of the corresponding GABARAPL1 variants, to further validate our biochemical and structural conclusions. In agreement with our previous data, K24E, L50A, and R67E from GABARAPL1 all abolished any interaction with LIR (Figure II). As suggested by the Reviewer, GABARAP R28 indeed was not involved in the interaction! Thanks for focusing our attention on this aspect; we have removed R28E variant data and have revised Fig.2c and Fig. 4b accordingly in our revised manuscript to clarify our results.

Figure II: ITC results showing that mutating the key residues in GABARAPL1 decrease the binding to γ 2-LIR when compared to the wild-type GABARAPL1. The error of K_d for each ITC curve represents the curve fitting error. N: stoichiometry of binding. The ‘N.D.’ denotes that there is no detectable binding.

2) The authors concluded that GABAA-R-gamma2 LIR is selective toward GABARAP and GABARAP-L1 among 6 homologs. Until now many structures have been reported for mammalian Atg8s complexed with various LIRs, which unveiled preference of LIRs for GABARAP and LC3 family. For example, GABARAP interaction motif (GIM) was proposed as a preferred sequence for GABARAP family (PMID 30518659). Compare the determined structure with previously reported structures and discuss the mechanism of how GABAA-R-gamma2 LIR selectivity binds GABARAP and GABARAP-L1 among six Atg8 homologs.

We thank the reviewer for these helpful suggestions. We have now compared several previously reported Atg8-LIR structures with our determined structure (Figure III) (including PDB: 5DPR, 5DPT, 5DPW, 5DPS from PMID 30518659, PDB: 3X0W from PMID 25498145, and PDB: 1EO6 from PMID 10856287; five of the structures are the complex formed with PLEKHM1 except that 1EO6 is the apo form of GABARAPL2). The RMSD values are summarized in the table below (Response Table I). From these comparisons, we can see that the overall folding of all the 6 homologs are very similar. Although our biochemical data showed that γ 2-LIR is in fact a GIM, the core motif of gamma2-LIR “YECL” does not match the preferred sequence of the GIM described by Rogov *et al.*¹. We therefore speculated that the preferred binding of γ 2-LIR to GABARAP/GABARAPL1 may result from other factors. Thus, we carefully re-analyzed the structures and the sequences of Atg8s (Figure IV). First, we noticed that E402 from γ 2-LIR form charge-charge interaction with K24 and hydrogen bonding with Y25. However, in LC3A/B/C, this “KY” is replaced by “QH” or “KF”. Indeed, mutating E402 to Thr decreases the binding by 2-fold and mutating K24 to Glu largely weakens the binding (Fig 2e and Figure I). In addition, Y405 interact with K48. It seems likely that the positioning of K48 is dependent upon hydrogen bonding with Y5. In GABARAPL2 and LC3A/B/C, this position is Phe. We have shown in our previous study that Y5 is functionally impactful for selective binding of AnkG-LIR². Therefore, our current view is that the specific binding between γ 2-GABA_AR and GABARAP/GABARAPL1 involves collective effects from each of these factors.

Figure III: Comparison of GABARAPL1/ γ 2-GIM with previously reported structures of Atg8s. GABARAPL1 in our structure is shown in green, and γ 2-GIM is shown in orange.

Response Table I: The RMSD values of GABARAPL1/ γ 2-GIM vs previously reported structures of Atg8s.

vs	GABARAP (5DPS)	GABARAPL1 (5DPT)	GABARAPL2 (1EO6)	LC3A (5DPR)	LC3B (3X0W)	LC3C (5DPW)
RMSD (Å)	0.545	0.540	0.731	0.789	0.990	1.182

Figure IV: Sequence alignment of six Atg8 members from human. The residues selected for mutagenesis or specificity analyses shown in this paper are indicated with cyan dots.

3) By the observation that the treatment of endocytosis inhibitor dynasore caused similar increase of GABA MCD for cells with and without expression of both GABA-A-R and GABARAP, the authors concluded that binding of GABARAP to GABA-A-R promotes the trafficking of the receptor from Golgi to cell membrane rather than inhibiting endocytosis. However, in this study the authors did not confirm that dynasore treatment alone blocked endocytosis sufficiently, and so there remains a possibility that endocytosis was partially inhibited by the expression of GABA-A-R and GABARAP, which was further inhibited by dynasore treatment. Perform back-up experiments, for example confirming the sufficient endocytosis inhibition by dynasore treatment or performing similar experiments using the cells AP2 is knocked-down.

We thank the reviewer for this guidance. Following the suggestions, we have now performed a parallel experiment using a higher concentration of dynasore (120 μM for 2h) (Figure V). Compared with the 80 μM dynasore concentration used in our original manuscript, the higher concentration of dynasore does not further elevate the GABA MCD (Figure V). These new results support that the dynasore concentration used in our assays does work as intended in our experimental design: it blocks endocytosis to a sufficient extent to evaluate the GABARAP's functional roles in promoting GABA_AR membrane localization.

Figure V: Effects of multiple dynasore concentrations on GABA_AR-mediated current densities in HEK-293 cells. Representative trace records and average values of GABA currents activated by 1 mM GABA in HEK-293 cells expressing GABA_AR ($\alpha 1\beta 2\gamma 2$) with or without treatment of the endocytosis inhibitor dynasore (80 μM or 120 μM , 2h). $n=8-10$. Data are represented as the mean \pm SEM. ** $P < 0.01$, based on One-Way ANOVA; ns, not significant ($P > 0.05$).

To further support our conclusion that GABARAP functions to promote the trafficking of the receptor, we have now performed experiments that enabled direct evaluation of the effects of GABARAP on receptor trafficking by treatment with Brefeldin A (BFA), which induces Golgi disassembly and is widely used to block the trafficking of membrane proteins^{3,4}. Our results showed that BFA treatment (5 $\mu\text{g}/\text{mL}$, 30 min) alone could reduce the GABA current densities, suggesting a decrease in GABA_AR membrane localization (Figure VI). Further, BFA treatment blocked the GABARAP-mediated increase in GABA currents as induced by 1 mM GABA (Figure VI), a finding demonstrating that GABARAP promotes receptor membrane localization via a trafficking pathway susceptibly regulated by BFA.

Figure VI: Effects of Brefeldin A (BFA) on GABA_AR-mediated current densities in HEK-293 cells. Representative trace records and average values of GABA currents activated by 1 mM GABA in HEK-293 cells co-expressing GABA_AR ($\alpha 1\beta 2\gamma 2$) and WT GABARAP with or without treatment of the protein trafficking inhibitor Brefeldin A (5 $\mu\text{g}/\text{mL}$, 30 min). $n=9-14$. Data are represented as the mean \pm SEM. * $P < 0.05$, *** $P < 0.001$ based on One-Way ANOVA; ns, not significant ($P > 0.05$).

Minor points

1) In Fig. 1b, legend, “gamma2-ICD is highlighted in red”, red is not used in the figure.

We thank the reviewer for pointing out our error in the original manuscript; we have now corrected this in the revised legend.

2) Provide raw data and N (stoichiometry of binding) values for all ITC data (Fig. 1c, d, 2d, e, 3b, d, e, 5a, b, Supple Table2).

Following the reviewer’s suggestion, we have added all of the raw data and labeled the corresponding N value for each of our ITC data presented below (ITC Response Figures 1-6). To make our manuscript concise, we did not put all the raw ITC data in our revised manuscript.

ITC Response Figure-1 (supplement to Fig. 1b, c): ITC-based mapping of the minimal GABARAP binding region in the $\gamma 2$ -GABA_AR. The minimal and complete GABARAP binding region identified is 398-415. The error of K_d for each ITC curve represents the curve fitting error. N: stoichiometry of binding. The 'N.D.' denotes that there is no detectable binding.

ITC Response Figure-2 (supplement to Fig. 1e): ITC results showing that $\gamma 2$ -LIR specifically binds to GABARAP and GABARAPL1 but none of the other Atg8 family members. The error of K_d for each ITC curve represents the curve fitting error. N: stoichiometry of binding. The 'N.D.' denotes that there is no detectable binding.

ITC Response Figure-3 (supplement to Fig. 2d): ITC results showing that mutating the key residues in GABARAP decrease the binding to γ 2-LIR when compared to the wild-type GABARAP. The error of K_d for each ITC curve represents the curve fitting error. N: stoichiometry of binding. The ‘N.D.’ denotes that there is no detectable binding.

ITC Response Figure-4 (supplement to Fig. 2e, Fig. 3b and Supplementary Table 2): ITC results comparing the binding affinities between GABARAP or AP2 and γ 2-LIR or its 5 mutant variants (E402T, Y403E, Y405E, E406R, and L408Q). The error of K_d for each ITC curve represents the curve fitting error. N: stoichiometry of binding. The ‘N.D.’ denotes that there is no detectable binding.

ITC Response Figure-5 (supplement to Fig. 6a, b): ITC results comparing the binding affinities between GABARAP or AP2 and a synthesized γ 2-LIR WT peptide or 3 phospho-peptides (pY403, pY405, and pY403pY405). The error of K_d for each ITC curve represents the curve fitting error. N: stoichiometry of binding. The ‘N.D.’ denotes that there is no detectable binding.

ITC Response Figure-6 (supplement to Fig. 7a, b): ITC results showing that giant ankyrin peptides (AnkB-WT, AnkG-WT, AnkG-ER) can strongly bind to GABARAP and AnkG-WT peptide can effectively disrupt the GABARAP/ γ 2-LIR interaction. The error of K_d for each ITC curve represents the curve fitting error. N: stoichiometry of binding. The ‘N.D.’ denotes that there is no detectable binding.

We would like to take this opportunity to again thank the reviewer for the excellent guidance about how to improve the technical rigour and scientific contribution of our study. Many thanks.

Reviewer #2 (Remarks to the Author):

This study outlines some new perspectives on the association of GABARAP with the GABAA receptor and identifies a new binding motif, which overlaps with the AP2 protein responsible for receptor internalisation. It is proposed that the competition between these proteins, which interact with the intracellular loop of gamma2 subunits, dictates the stability of the membrane bound receptor, and provides a plasticity 'handle'. Besides binding affinity data, crystallographic evidence is presented for the GABARAP interaction, as well as functional experiments that are used to establish the cellular role of GABARAP in the GABAA receptor's ability to access and remain in the cell membrane. This is a good but preliminary study. It re-addresses potential oversights in the association between two key proteins which affect neuronal activity, but there are a significant number of major issues which detract from the study's quality.

1. The structural evidence for a novel binding site between GABARAP and the gamma2 loop (G2-LIR) seems plausible, though the three key observations used from the literature to question current opinion on the role of GABARAP are selective and arguably tenuous. That said, they do provide a reason for this study. However, the general conclusions pertaining to GABARAP's neuronal function do not differ much from those already published. The more novel aspect of the paper, that exploring the interaction and competition between GABARAP and AP2, and the potential to tune receptor levels in the membrane, potentially via phosphorylation, and affect plasticity, is insufficiently explored. I have highlighted many questionable areas of the study below.

We thank the reviewer for these comments and appreciate the recognition of our novel binding site identification and competition between GABARAP and AP2 with G2-LIR. The long-term focus of our research program has been on Ankyrin family proteins. While investigating the super-strong binding between Ankyrins and GABARAPs, we came across the current observations that launched the present study about GABARAP-GABA_AR interactions. In planning how to investigate the functional contributions of ankyrins on this biologically critical complex, our literature searching focused our attention on the physiological process of GABA_AR membrane localization and how it is regulated by GABARAP. We have now carefully modified the introduction section of our revised manuscript to accurately present current knowledge about GABARAP's functional roles in regulating GABA_AR membrane localization. Happily, we have now completed a large set of diverse experiments that address the carefully considered concerns offered by the reviewer.

2. Interaction site/structure

Crystal structures for GABARAP, AP2 and the gamma2 LIR (in various combinations) already exist. This detracts from the impact of the crystallographic data, although breakdown of the key residues which participate in the GABARAP:G2-LIR interaction (and their subsequent mutational evidence) is interesting and novel. The previously reported G2-LIR:AP2 complex crystal structure (which is incorrectly cited in the text) has much overlap with this study in reporting a similar interaction site.

The Atg8 family of proteins (of which GABARAP is a member) have 'high sequence similarity'

yet only two bind to the G2-LIR. It would be useful to have more information as to why this is. Notably, the L1 variant of GABARAP binds equally well, but reasoning as to why the native (and more physiologically relevant) GABARAP was not crystallized is absent. We need to be better informed about the degree of sequence similarity between these variants, as there is total reliance on L1 for all of the structural interpretation, not only with LIR, but also with AP2 binding. Given this reliance, it needs to be established if GABARAP and GABARAPL1 have similar effects in the key functional e-phys experiments. If these controls are not present, then the validation of the G2-LIR:GABARAPL1 X-stal model is questionable.

Also, whilst performing the crystallizations, it would have been prescient to co-crystallize the previously reported 424-441 peptide, to verify non-binding in this different format.

We thank the reviewer for appreciating the value of our structure for identifying the key residues participating in the GABARAP:G2-LIR interaction. Indeed, the structures of the GABARAP apo form, AP2 apo form, and AP2:G2-LIR complex have already been reported. However, no GABARAP:G2-LIR complex structure was available prior to our study, although GABARAP was first identified as the binding partner for G2-GABA_AR. The pioneer study by Wang H, et al⁵ and a series of excellent follow-up studies have established the general functional role of GABARAP in GABA_AR at the cellular and physiological levels. However, as structural biologists, our interests are of course focused on understanding how GABA_AR binds to GABARAP at the atomic level. Given the high sequence similarity between GABARAP and GABARAPL1 (sequence identity 87.1%, Figure IV), as well as its similar binding affinity with G2-LIR (K_d : 4.9 μ M vs 6.6 μ M), we actually tried crystallization of both GABARAP:G2-LIR and GABARAPL1:G2-LIR. However, GABARAP:G2-LIR yielded no crystals. The only high-resolution structure we got is GABARAPL1:G2-LIR. This structure was, nevertheless, highly informative, and launched our extensive experimental investigations in our follow-up biochemical assays. We now have extensive additional experimental evidence to lend support to our best current interpretations about the GABARAP–GABA_AR binding mode. We now have reviewer-suggested ITC & GST-pull down data, and these results validate the functions of the key binding residues we observed in our crystal structure; note that we have such data for both GABARAP and GABARAPL1 (Figure I and II in the response to Reviewer#1's point-1, and see details below).

Regarding the specific binding between G2-LIR and 6 Atg8 members (also raised by the Reviewer#1), we have now compared several previously reported Atg8-LIR structures with our determined structure (Figure III), including (PDB: 5DPR, 5DPT, 5DPW, 5DPS from PMID 30518659, PDB: 3X0W from PMID 25498145, and PDB: 1EO6 from PMID 10856287; five of the structures are the complex formed with PLEKHM1 except that 1EO6 is the apo form of GABARAPL2). The RMSD values are summarized in the table below (Response Table I). From these comparisons, we can see that the overall folding of all the 6 homologs are very similar. Although our biochemical data showed that γ 2-LIR is in fact a GIM, the core motif of gamma2-LIR "YECL" does not match the preferred sequence of the GIM described by Rogov *et al.*¹. We therefore speculated that the preferred binding of γ 2-LIR to GABARAP/GABARAPL1 may results from other factors. Thus, we carefully re-analyzed the structures and the sequences of Atg8s (Figure IV). First, we noticed that E402 from γ 2-LIR form charge-charge interaction with K24 and hydrogen bonding with Y25. However, in LC3A/B/C, this "KY" is replaced by "QH" or

“KF”. Indeed, mutating E402 to Thr decreases the binding by 2-fold and mutating K24 to Glu largely weakens the binding (Fig 2e and Figure I). In addition, Y405 interact with K48. It seems likely that the positioning of K48 is dependent upon hydrogen bonding with Y5. In GABARAPL2 and LC3A/B/C, this position is Phe. We have shown in our previous study² that Y5 is functionally impactful for selective binding of AnkG-LIR². Therefore, our current view is that the specific binding between γ 2-GABA_AR and GABARAP/GABARAPL1 involves collective effects from each of these factors.

Figure III: Comparison of GABARAPL1/ γ 2-GIM with previously reported structures of Atg8s. GABARAPL1 in our structure is shown in green, and γ 2-GIM is shown in orange.

Response Table I: The RMSD values of GABARAPL1/ γ 2-GIM vs previously reported structures of Atg8s.

vs	GABARAP (5DPS)	GABARAPL1 (5DPT)	GABARAPL2 (1EO6)	LC3A (5DPR)	LC3B (3X0W)	LC3C (5DPW)
RMSD (Å)	0.545	0.540	0.731	0.789	0.990	1.182

Figure IV: Sequence alignment of six Atg8 members from human. The residues selected for mutagenesis or specificity analyses shown in this paper are indicated with cyan dots.

As suggested by the reviewer, to further validate the G2-LIR:GABARAPL1 X-stal model, we have performed experiments showing that GABARAPL1 has similar effect with GABARAP on the GABA currents in our revised manuscript (Figure VII). Collectively, these data demonstrate that GABARAP and GABARAPL1 are apparently indistinguishable with regard to GABA_AR binding at the structural, biochemical, and electrophysiological levels.

Figure VII: Effects of GABARAP and GABARAPL1 on GABA_AR-mediated current densities in HEK-293 cells. Representative trace records and average values of GABA currents activated by 1 mM GABA in HEK-293 cells co-expressing GABA_AR (α1β2γ2) or GABA_AR (α1β2γ2^{E406R}) and GABARAP or GABARAPL1. n=10–13. Data are represented as the mean ± SEM. *** *P* < 0.001 based on One-Way ANOVA; ns, not significant (*P* > 0.05).

Further, our observation of no apparent binding of GABARAP with the previously reported 424–

441 peptide (*Nota bene*, using our experimental conditions) indicate that this peptide may not be associated with GABARAP. Given the requirement for a protein-protein interaction to achieve co-crystallization, and considering the lack of any detected G2:424–441 peptide interaction, it was not possible to conduct structural analysis of such a complex.

3. Pull-downs and binding

There is an over-reliance on the ITC technique (and analytical gel filtration), which is used throughout the study, for establishing the binding interaction, and it may present interpretational challenges. The ITC technique is sensitive, but the LIR peptide is derived from a membrane-bound protein so its structure in solution may be artificial. The authors will know that many previous studies have very successfully used GST pull-down assays to study this interaction. This latter technique should be more prominent here with regard to quantifying binding of the novel protein motif/peptide, as the initial starting structures for the peptide(s) from these techniques may vary more than anticipated. It will satisfy readers who feel that abject dismissal of the previously published peptide as a binding motif is premature. It would be useful to have ITC calorimetry data for the full length isolated protein (membrane-bound?) for comparison to the isolated peptide.

Thanks for the helpful guidance about how to properly interpret the data using a peptide derived from a membrane protein. The LIR located in the middle of the long intracellular loop between transmembrane helices TM3 and TM4 (aa 375-442). This loop was previously characterized as “unstructured”, a conclusion which was very recently confirmed by several high-resolution cryo-EM structures (*e.g.*, PDB: 6HUO)⁶. We have now compared the conformations of G2-LIR in complex with GABARAP or AP2 by superimposing the “YECL” motif (Figure VIII). These comparisons highlighted that the peptide conformations are quite different upon binding with different targets; one interpretation here is that the peptide itself may not adopt a fixed conformation; rather its structure could be induced upon its binding to a specific target.

Figure VIII: Compare the conformations of G2-LIR in complex with GABARAP or AP2 by superimposing the “YECL” motif.

As suggested by the reviewer, and to further confirm the binding using the full length protein, we have performed GST pull-down assays in various combinations. We show that purified GST-GABARAP can successfully pull-down full length GFP- γ 2-GABA_AR WT protein expressed in HEK-293 cells. In contrast, binding was abolished between the full length receptor with a LIR 398-415 deletion or a E406R variant. However, GABARAP can still interact with GABA_AR with a 424-441 deletion, a finding indicating that this previously reported motif may not participate in binding (Figure IX).

Figure IX: GST-pull down assays showing that the fragment 398-415 of γ 2-GABA_AR is responsible for GABARAP binding; besides, key residue γ 2-GABA_AR E406 involved in the GABARAP/ γ 2 398-415 interface is required for the intact interaction.

We also utilized GST pull-down assays to verify the binding specificity with diverse Atg8 members. Our results showed that GST-GABARAP and GST-GABARAPL1 bind to GFP- γ 2-GABA_AR with nearly equal affinity, whereas no binding was detected for GST-LC3A or a GST-GABARAP L50A variant (Figure X). Regarding the ITC assay with the purified full length protein: there is a major technical challenge here; the need to purify a full length membrane protein, and to do so with adequate yield to perform meaningful ITC experiments, is beyond the capacities of our lab, those of any of our collaborators or colleagues, and any commercial research operations.

Figure X: GST-pull down assays showing that γ 2-GABA_AR FL specifically binds to GABARAP and GABARAPL1 but not LC3A; besides, key residue of GABARAP L50 is required for the GABARAP/ γ 2 interaction.

4. G2-LIR mutations

Please give some reasoning behind the selected mutations for this peptide, as this is unclear. Very few seem to conserve charge (eg R28E), whilst others introduce charge (which may explain the increase in binding with Y403E). The appropriateness of the mutations needs rationalising - were just key (interacting?) residues mutated, or was this extended to other/all residues in the peptide. Also, did the K_d remain unchanged for supposedly unimportant residue mutation (see below)?

Given that the G2-LIR peptide is only 18aa long, some mutations may affect the overall backbone of the peptide, and could be the reason why some mutations are more/less effective than others. Some comment about this is needed.

Thanks for the suggestion. We apologize for not explaining the rationale of the mutations well. All the selected mutations are the key residues we identified from the structure (except R28, which we have now removed from the revised manuscript in light of new evidence from experiments directed by Reviewer#1; for specifics on this residue, kindly see our response to point-1 of Reviewer#1). Some of the residues function in GABARAP-GABA_AR binding through their sidechains. For example, our structure indicates that negatively charged residue E406 from G2-LIR interacts with the positively charged residue R67 from GABARAP. We therefore anticipated that a charge reversal mutation (*i.e.*, mutating the negatively charged E406 to Arg) may i) disrupt the charge-charge interaction and ii) potentially introduce a charge repulsion effect; this would strongly impair binding; indeed, this was confirmed in our binding data. Similarly, our structure indicated that the Y405 and L408 residues from G2-LIR participate in GABARAP-GABA_AR binding through the positioning of their sidechains inserted into hydrophobic pockets. Thus, on the one hand we anticipated that mutating Y405 or L408 to polar residues Glu or Gln should weaken binding (note that our decision to mutate Y405 to Glu is because this is also a phosphomimic mutation). Again, our binding assays with these variants support our speculations based on our complex structure. On the other hand, we anticipated that mutating GABARAP Y49 or L50 to Ala would change the shape of the two hydrophobic pockets; these mutations also abolished the binding. E402T mutation is to explain why GABARAP prefers binding to gamma2 subunit but not gamma3⁷, because this position is a Thr in human gamma3 subunit. We have added these explications for each selected mutation in the revised manuscript. We have also added both biochemical and e-phys data about several other key residues from the binding interface (Fig. 2c,d, and Fig. 4b in our revised manuscript), as also guided by Reviewer#1 (for this detailed content, kindly see our responses to point-1 of Reviewer#1).

As we discussed in the response to point-3, our current model for the binding mode for the peptide is that the peptide does not manifest in any fixed structure in the absence of a binding partner; rather both the backbone and sidechain conformations of the residues in this peptide are determined upon binding to its specific target. We did not introduce any residues (Pro or Gly) that have discrete dihedral angles profiles, so our data support that the binding-related impacts of the examined mutations result from sidechain contributions.

5. Main functional data.

The conclusions from this data largely echo those currently in the literature (whilst noting the novelty of the binding of the LIR(398-415) peptide), i.e. the role of GABARAP is unchanged. Much of the data to get to this point is still quite preliminary. The co-expression studies need to have more controls – i.e. concentration-response curves to establish that there are no changes in GABA potency and/or GABA efficacy in the presence of the peptides. There needs to be at least some basic level of immuno-cytochemistry in place to verify increases in receptor number in both HEK cells and neurons, and checks with Golgi markers to quantitate if trafficking is facilitated (or not) in the presence of GABARAP and the peptides.

We thank the reviewer for this helpful guidance. We have now verified the effect of GABARAP on GABA potency by examining concentration-response curves in HEK-293 cells co-expressing GABA_AR and GABARAP. Although the presence of GABARAP significantly increased GABA-induced maximum current, this did not alter its EC₅₀ value for GABA (Figure XI). These findings support that GABARAP does not apparently impact the potency of GABA on GABA_AR.

Figure XI: Effects of GABARAP on the efficacy of GABA on GABA_AR in HEK-293 cells. **a**, Dose-response curves of I_{GABA} in HEK-293 cells co-expressing GABA_AR α 1 β 2 γ 2 and GABARAP. The data were normalized to I_{max} of the GABA_AR+Vector group. GABA at 1 μ M, 3 μ M, 10 μ M, 30 μ M, 100 μ M, 300 μ M and 1000 μ M were selected. $n=7$. **b**, EC₅₀ values of I_{GABA} induced by increasing GABA concentrations in HEK-293 cells co-expressing GABA_AR (α 1 β 2 γ 2) and GABARAP. $n=7$. Data are represented as the mean \pm SEM. ns, not significant ($P > 0.05$) based on unpaired t test.

We have tried several times to image the membrane-localized GABA_AR by immuno-staining using fluorescence microscopy. However, due to some technique issues and maybe the nature of GABA_AR, we have not yet obtained a clear immunostaining signal. Thus, we cannot verify increases in receptor number in HEK cells or neurons through immunostaining, nor checking with Golgi markers. Alternatively, we have performed cell surface biotinylation analysis of GABA_AR in HEK-293 cells with or without GABARAP WT (or mutation variants). This biochemical assay has been widely used to quantify specific membrane-bound proteins. Our results show that both GABARAP and GABARAPL1 can increase number of receptors at the membrane surface; moreover, such increases do not occur with binding-deficient mutation variants of GABARAP (L50A or R67E) or of GABA_AR (E406R) (Figure XII).

Figure XII: Surface biotinylation analysis of the GABA_ARs. HEK-293 cells co-expressing GABA_AR ($\alpha 1\beta 2\gamma 2$) or GABA_AR ($\alpha 1\beta 2\gamma 2^{E406R}$) and WT GABARAP or GABARAPL1 or mutant GABARAP (L50A and R67E were selected) followed by cell surface biotinylation/immunoblotting. Data are represented as the mean \pm SEM (n=3). ** $P < 0.01$ based on unpaired t test; ns, not significant ($P > 0.05$).

We also evaluate the effects of Ankyrin-B or LIR peptide from GABA_AR on GABA_AR membrane surface localization (Figure XIII): Ankyrin-B or LIR alone cannot promote increase of surface amount (Figure XIII lane 4 and 5). Interestingly, G2-LIR can block the GABARAP-mediated increase of GABA_AR membrane surface localization (Figure XIII lane 6), which is consistent with our results using e-phys approach (Figure XVIIIa, and see details in our response to point-7 below).

Figure XIII: Surface biotinylation analysis of the GABA_A receptor. HEK-293 cells co-expressing GABA_AR (α 1 β 2 γ 2) and GABARAP or GABARAPL1, AnkB-WT peptide, γ 2 398-415 peptide followed by cell surface biotinylation/immunoblotting. Data are represented as the mean \pm SEM (n=3). * $P < 0.05$, ** $P < 0.01$ based on unpaired t test; ns, not significant ($P > 0.05$).

Cell recordings are made using a maximal dose of GABA on cells which likely express α 1 β 2 γ 2 receptors very efficiently. If these cells are not compensated for series resistance (not reported), they will be prone to underestimation of the maximal currents, which is the only metric used in these experiments. Not every GABA-A receptor in the neuron is the same, so co-expression with an extrasynaptic receptor type should be included for a few of the conditions.

Thanks for the suggestion. We are sorry for the lack of sufficient descriptions for our series resistance experiments. Actually, as we now carefully detail in a reworked subsection of the revised methods, we did properly compensate for the series resistance for all of the recorded HEK-293 cells in the experiments of the originally submitted manuscript. "Series resistance compensation of 60–80% was used and monitored in the whole-cell recording experiments".

We selected α 5-containing GABA_ARs (α 5 β 2 γ 2) as a representative for an extrasynaptic GABA_AR^{8,9}, and examined the effect of GABARAP on this type of GABA_AR. GABARAP also significantly increased the GABA_AR-mediated current as induced by 1mM GABA (Figure XIV). These results suggest that γ 2-containing extrasynaptic GABA_ARs are also sensitive to regulation by GABARAP; this conclusion further supports our observation that GABARAP increases the membrane GABA_AR level and GABA-induced current, resulting from its interaction with γ 2 subunits.

Figure XIV. Effects of GABARAP on GABA_AR-mediated current densities in HEK-293 cells co-expressing GABARAP and α 5-containing GABA_ARs. Representative trace records and average values of maximum current density activated by 1 mM GABA in HEK-293 cells co-expressing GABA_ARs (α 5 β 2 γ 2) and WT GABARAP or GABARAP mutants. n=10–12. Data are represented as the mean \pm SEM. * P < 0.05, ** P < 0.01 based on One-Way ANOVA.

6. Phosphorylation

It is not clear how phosphorylated forms of Y403 & 5 are generated in the study. The result (line 272) that phosphorylated Y405 and Y403/5 means neither AP2 nor GABARAP bind the G2-LIR is puzzling. Given that these two residues are both tyrosines so are likely to be equally susceptible to phosphorylation/de-phosphorylation by kinases/phosphatases, it is hard to reconcile how only phosphorylation of Y403 (not Y405) is likely to happen in neurons to permit GABARAP/AP2 binding. These phosphorylation experiments (not simply phospho-dead/mimetic mutations, but also kinase supplementation experiments) need to be undertaken in the functional ephys assays to understand the complete effects of these changes, and to give them a more relevant context.

Thank the reviewer for this suggestion. We also think tyrosine phosphorylation seems to be a plausible mechanism regulating the binding of GABA_AR with GABARAP vs AP2 and quite interesting to be evaluated. We are sorry for the undetailed description of the peptides used in our study. The phosphorylated forms of Y403, Y405, and double-phosphorylation of Y403/Y405 of LIR peptides (Fig. 3d,e in our original manuscript) are commercially synthesized and used for ITC experiments. The purity and molecular weight were verified by HPLC and mass spectrometry from the commercial research operations. Our ITC results showed that phosphorylated Y405 and Y403/Y405 double phosphorylation could not bind with neither GABARAP nor AP2, which is highly expectable from the complex crystal structures. We have now added more relevant context in our revised manuscript guided by the reviewer. We agree that it may be quite complex *in vivo* and how only one of the two tyrosines is phosphorylated is currently unknown. It is also technically challenging to distinct the phosphorylation status for the two residues adjacent both *in vitro* and *in vivo*.

As suggested by the reviewer, we have performed the kinase supplementation experiments by co-expression with the Src kinase (which was reported to phosphorylate the two tyrosine residues of γ 2-GABA_AR)^{10,11} and with or without further treatment by Src kinase inhibitor PP2¹² in our electrophysiological assays. Our results showed that co-expression with Src kinase blocked any GABARAP-mediated increase in GABA currents as induced by 1 mM GABA (Figure XVI). However, further treatment with PP2 restored the increase effect of GABARAP on GABA

currents (Figure XVI), indicating PP2 treatment blocked Src kinase activity sufficiently. These results suggested that phosphorylation of γ 2-GABA_AR indeed could regulate its membrane localization.

Figure XVI: Effects of Src kinase on GABA_AR-mediated current densities in HEK-293 cells co-expressing GABARAP. Representative trace records and average values of GABA currents activated by 1 mM GABA in HEK-293 cells co-expressing GABA_AR (α 1 β 2 γ 2), GABARAP and Src kinase with or without treatment of kinase inhibitor PP2. n=9–10. Data are represented as the mean \pm SEM. *** $P < 0.001$ based on One-Way ANOVA; ns, not significant ($P > 0.05$).

7. Ankyrins

The competition between GABARAP & AP2 for binding sites, and their functional consequences, are important to understand for this study. However, this interesting concept is completely overlooked. Instead the authors opt to disrupt the G2-LIR:GABARAP interaction site with ankyrins. These attachment proteins were chosen because the modest binding of G2-LIR peptide (as a competitive inhibitor) was deemed to be a ‘sub-optimal tool’. This competition experiment should have at least have been tried (even in HEK cells). The ankyrins chosen instead disrupt GABARAP binding much more effectively. However, we are not told where they bind, as, if it is not overlapping with the LIR site, then they are not wholly appropriate. Also, cortical neurons are used to test the effectiveness (and physiological consequences) of disrupting the GABAA-R:GABARAP interaction, yet this interaction needs controls to be performed in HEK cells (at least) before their consequences can be properly interpreted in neurons following viral transfections. Do ankyrins even affect the behaviour of GABAA-Rs in HEK cells in the absence of GABARAP? Generally, the experiments associated with figure 5 are poorly controlled.

We thank the reviewer for this helpful guidance. We have performed experiments to show that ankyrin-G competes with GABA_AR in binding to GABARAP (Fig. 7a,b). We showed that GABA_AR could not bind to GABARAP in the present of ankyrin-G (AnkG) through ITC assay in our original manuscript (Fig. 7b blue line). As suggested by the reviewer, we have now performed GST pull-down experiment showing that ankyrin-B (AnkB) peptide can also disrupt the GABARAP/ γ 2-GABA_AR interaction (Figure XVII). We have added the description that ankyrins bind to the overlapping region on GABARAP with the LIR interaction sites in our revised manuscript.

Figure XVII: GST pull-down assay showing that the AnkB WT peptide can effectively disrupt the GABARAP/ γ 2-GABA_AR interaction.

We have now performed the competition experiment using G2-LIR in HEK-293 cells guided by the reviewer. Our new experiments showed that overexpression of G2-LIR could indeed eliminate the GABARAP-mediated increase in GABA currents (Figure XVIIIa). Consistent with our biochemical data, AnkB-WT peptide also blocked GABARAP-mediated increase in GABA currents (Figure XVIIIa), while AnkB-WT peptide itself had no effect on the GABA currents in the absence of GABARAP (Figure XVIIIb).

Figure XVIII: Effects of G2-LIR and AnkB-WT on GABA_AR-mediated current densities in HEK-293 cells. **a**, Representative trace records and average values of GABA currents activated by 1 mM GABA in HEK-293 cells co-expressing GABA_AR (α 1 β 2 γ 2), GABARAP and G2-LIR or AnkB-WT. $n=10-12$. Data are represented as the mean \pm SEM. *** $P < 0.001$ based on One-Way ANOVA; ns, not significant ($P > 0.05$). **b**, Representative trace records and average values of GABA currents activated by 1 mM GABA in HEK-293 cells co-expressing GABA_AR (α 1 β 2 γ 2) and AnkB-WT. $n=10$. Data are represented as the mean \pm SEM. ns, not significant ($P > 0.05$).

8. Methods, n numbers, statistics, figures

Many of the figures/data are presented as if they are an $n=1$. Clearly more than one experiment has been performed in most cases as Kd values are \pm errors. The text, legends and methods have no (or very few) N numbers reported, and no statistical analysis, so we cannot assess how robust the results are.

We thank the reviewer for the helpful guidance. We are sorry for the lack of sufficient descriptions

about the methods, n numbers and statistics in our original manuscript. We have now carefully re-examined and modified these corresponding parts in the revised manuscript.

Many of the supplementary data figures will be meaningless to many readers. The fluorescence assay (Sfig1) is difficult to interpret for quantitation. Similarly for sFig3 & 4 chromatography plots. Both these assays are accompanied by insubstantial Methods sections which lack the necessary detail expected of the journal, and do not help interpretation of these figures. The Methods section on the crystallography also falls far short on detail that is expected, and many other Methods sections are similarly lacking.

Thanks for these suggestions. We have now re-organized both the main figures and the supplementary figures with input of our revised experimental results. Particularly, we have removed the fluorescence assay figure as suggested by the reviewer. We have added corresponding descriptions that could help to interpret our supplementary data about chromatography plots and crystallography.

Figure 3a does not clearly represent what is being discussed in the text. This peptide complex structure (PDB code 2PR9) needs to be cited. I don't think this is a model generated by this study (it is not clear), so this citation needs to be present. If the model is derived here, then no docking methodology is included in the text. This data's origin is confusingly presented.

Sorry for the confusion, Figure 3a is generated using the previous reported peptide complex structure (PDB code 2PR9), not a model generated in this study. We have used the PDB file of PDB: 2PR9 and the software PyMOL to create a figure showing the binding interface between G2-LIR and AP2. We have now modified Figure 3a and added clear citation label in the figure.

The PDB code for the new Xstal complex needs to appear in the Methods (as well as fig legend). Thanks for the note, and we have made the suggested change in the revised manuscript.

Fig 5f states that the treatments do not change frequency. This is the only figure where any statistics are presented yet the foundation for this is based on data which changes in frequency from <1 Hz to 20 Hz. The Methods do not say at what age slice recording was happening, so this variability (which needs scrutiny) may result from very different age animals.

Thanks for the kind guidance. We have added the age of the mice used in our study in the revised manuscript. We have checked the data in detail and confirmed that the variance in the mIPSC frequency is indeed existed. The large variance of mIPSC frequency has also been reported in several previous studies^{13,14}. Such case may be attributed to the different amounts of synaptic contacts with each neuron, the different synaptic damage during the processing of brain slice cutting, and other unknown reasons. We have provided some previously published examples as representative as following (Figure XIX).

Figure XIX: Representative examples showing the variance of the frequency of mIPSC from previous reported study¹³.

Some labelling in sFig2 is not correct. The gel in sFig4b looks overloaded or over-exposed?

Thanks for the note, and we have corrected the labelling in the revised manuscript. We carefully re-examined the gel in sFig4b and we think it should be fine.

We would like to take this opportunity to again thank the reviewer for the excellent guidance about how to improve the technical rigour and scientific contribution of our study. Many thanks.

Reviewer #3 (Remarks to the Author):

The manuscript by Jin Ye et al. describes the structure of the complex between GABARAPL1 and a peptide corresponding to the GABARAP-interacting region of γ 2-GABAAR. It also reports extensive complementary experiments supporting the structural findings and describing the details of the GABARAP/ γ 2-GABAAR interactions in vivo and in vitro. The experiments are well designed and well conducted; the data is compelling, interesting, and advances the understanding of how inhibitory synaptic transmission is regulated. Unfortunately the results are significantly overinterpreted. The most glaring examples are outlined below:

We thank the reviewer for the careful reading of our study and for appreciating the value of our work. We are also grateful for the extremely helpful guidance about how to improve the quality and purport of our study with additional confirmatory experiments. We have now addressed all the reviewer's concerns, and have revised our manuscript to incorporate our new data and the Reviewer-directed changes in emphasis.

- The authors claim in the Abstract, Results Section and Discussion that they “demonstrate that GABARAP functions to stabilize GABAARs via promoting its trafficking pathway instead of blocking receptor endocytosis”. This is not really the case. While the data is more consistent with GABARAP promoting GABAR trafficking, than with blocking receptor endocytosis, it does not prove the former, just suggests it. The facts that: 1: the endocytosis inhibitor dynasore blocks some of the receptor endocytosis (authors say previous studies show a ~90% block, but do not show how much it actually blocks in the assay they use), and 2: addition of GABARAP further increases the GABA MCDs, do not demonstrate that GABARAP does not block endocytosis. Indeed, the findings could also be explained by GABARAP blocking the remaining (say 10%) receptor endocytosis. The authors do not directly show that GABARAP increases receptor trafficking.

We thank the reviewer for this guidance. We agree with the reviewer that our original experiments did not fully support a trafficking role of GABARAP in promoting GABA_AR membrane localization; indeed, we cannot exclude the possibility that dynasore inhibition is not sufficient (also suggested by point-3 of Reviewer#1). Following the reviewer's suggestions, we have now performed a parallel experiment using a higher concentration of dynasore (120 μ M for 2h) (Figure V). Compared with the 80 μ M dynasore concentration used in our original manuscript, the higher concentration of dynasore does not further elevate the GABA MCD (Figure V). These new results support that the dynasore concentration used in our assays does work as intended in our experimental design: it blocks endocytosis to a sufficient extent to evaluate the GABARAP's functional roles in promoting GABA_AR membrane localization.

Figure V: Effects of multiple dynasore concentrations on GABA_AR-mediated current densities in HEK-293 cells. Representative trace records and average values of GABA currents activated by 1 mM GABA in HEK-293 cells expressing GABA_AR ($\alpha 1\beta 2\gamma 2$) with or without treatment of the endocytosis inhibitor dynasore (80 μ M or 120 μ M, 2h). n=8–10. Data are represented as the mean \pm SEM. ** $P < 0.01$, based on One-Way ANOVA; ns, not significant ($P > 0.05$).

To further support our conclusion that GABARAP functions to promote the trafficking of the receptor, we have now performed experiments that enabled direct evaluation of the effects of GABARAP on receptor trafficking by treatment with Brefeldin A (BFA), which induces Golgi disassembly and is widely used to block the trafficking of membrane proteins^{3,4}. Our results showed that BFA treatment (5 μ g/mL, 30 min) alone could reduce the GABA current densities, indicating a decrease in GABA_AR membrane localization (Figure VI). Further, BFA treatment blocked the GABARAP-mediated increase in GABA currents as induced by 1 mM GABA (Figure VI), a finding demonstrating that GABARAP promotes receptor membrane localization via a trafficking pathway susceptibly regulated by BFA.

Figure VI: Effects of Brefeldin A (BFA) on GABA_AR-mediated current densities in HEK-293 cells. Representative trace records and average values of GABA currents activated by 1 mM GABA in HEK-293 cells co-expressing GABA_AR ($\alpha 1\beta 2\gamma 2$) and WT GABARAP with or without treatment of the protein trafficking inhibitor Brefeldin A (5 μ g/mL, 30 min). n=9–14. Data are represented as the mean \pm SEM. * $P < 0.05$, *** $P < 0.001$ based on One-Way ANOVA; ns, not significant ($P > 0.05$).

- The model for GABARAP function that the authors propose in the Discussion Section (line 401 – till the end) is solely based on: “According to our results, giant ankyrin-G cannot form a triple complex with GABARAP and GABAAR, but rather competes out GABAAR by binding to the same region of GABARAP with super-strong affinity”. But the authors never actually show this in this manuscript. They only show that the ankyrin peptides reduce the mIPSC amplitudes and assume that this reduction is due to the peptides blocking the GABARAP/GABAAR interaction based on the previously published papers describing these peptides: line 329: “we conducted experiments using previously reported potent Atg8-targeting inhibitory peptides from giant ankyrins ... these peptides induce interactions which prevent the GABARAP and GABARAPL1 proteins from binding with GABAAR”. So everything that the authors discuss and propose starting in line 401 – till the end, is solely based on previously reported experiments and not on the data reported in this manuscript.

I suggest the authors revise their manuscript so that their conclusions and interpretations match the presented data.

We thank the reviewer for the suggestions. Actually, we only reported that giant ankyrins can potentially bind with Atg8 family members and served as inhibitory tools for autophagy field in our previous study². In that published paper, there are no any experimental results about GABA_AR. We discovered, in the present study, that GABARAP binds with GABA_AR through the overlapping region where ankyrins bind and that further guided us to evaluate the relationships among the three proteins. We have indeed performed experiments to show that ankyrin-G competes with GABA_AR in binding to GABARAP (Fig. 7a,b). We showed that GABA_AR could not bind to GABARAP in the present of ankyrin-G (AnkG) through ITC assay in our original manuscript (Fig. 7b blue line). We have now performed GST pull-down experiment showing that ankyrin-B (AnkB) peptide can also disrupt the GABARAP/γ2-GABA_AR interaction (Figure XVII). Collectively, our biochemical data in the revised manuscript have showed that giant ankyrins cannot form a triple complex with GABARAP and GABA_AR, but rather competes out GABA_AR. The novel crystal structure (GABARAP/GABA_AR) solved in this study, together with several crystal structures of GABARAP/ankyrins reported in previous study², showed that GABARAP utilized overlapping regions to bind with either GABA_AR or ankyrins. We agree with the reviewer that our tentative model wherein giant ankyrin-G may control the membrane anchoring of GABA_AR by releasing GABA_AR from GABARAP-bound trafficking complexes *in vivo* is not fully confirmatory. We have now deleted these parts and carefully modified the discussion section of our revised manuscript to accurately interpret our presented data as directed by the reviewer.

Figure XVII: GST pull-down assay showing that the AnkB WT peptide can effectively disrupt the GABARAP/γ2-GABA_AR interaction.

Minor points:

- The manuscript needs editing to improve the use of the English language (including use of commas, tenses, “the”/”a” etc.); for example what does “our results uncover the mechanism governing the binding site between GABA_AR and GABARAP” (line 46) mean? What is a mechanism that governs a site?

We thank the reviewer for pointing out these mistakes in our writing. We have corrected them in the revised manuscript as suggested by the reviewer. Furthermore, we have taken extra care to improve the writing of our manuscript.

- The authors have named their peptide γ 2-LIR for LC3-Interacting Region, but this peptide/region actually does not interact with any of the LC3s (line 174).

We thank the reviewer for pointing this out. Inspired by both Reviewer#3 and Reviewer#1, we have re-named our peptide γ 2-GIM (GABARAP interacting motif) instead of LIR (LC3 interacting region). We have changed all the corresponding parts. Thank the reviewer for the wonderful suggestion.

We would like to take this opportunity to again thank the reviewer for the excellent guidance about how to improve the technical rigour and scientific contribution of our study. Many thanks.

References:

1. Rogov VV. et al. Structural and functional analysis of the GABARAP interaction motif (GIM). *EMBO Rep* **18**, 1382-1396 (2017).
2. Li, J. et al. Potent and specific Atg8-targeting autophagy inhibitory peptides from giant ankyrins. *Nat Chem Biol* **14**, 778-787 (2018).
3. Chardin P, McCormick F. Brefeldin A: the advantage of being uncompetitive. *Cell* **97**, 153-5 (1999).
4. Su, Y.Y. et al. KIF5B Promotes the Forward Transport and Axonal Function of the Voltage-Gated Sodium Channel Nav1.8. *Journal of Neuroscience* **33**, 17884-17896 (2013).
5. Wang H, Bedford FK, Brandon NJ, Moss SJ, Olsen RW. GABA(A)-receptor-associated protein links GABA(A) receptors and the cytoskeleton. *Nature*. **397**, 69-72 (1999).
6. Masiulis, S. et al. GABAA receptor signalling mechanisms revealed by structural pharmacology. *Nature* **565**, 454-459 (2019).
7. Nymann-Andersen J, Wang H, Chen L, Kittler JT, Moss SJ, Olsen RW. Subunit specificity and interaction domain between GABA(A) receptor-associated protein (GABARAP) and GABA(A) receptors. *J Neurochem* **80**, 815-23 (2002).
8. Brickley S & Mody, I. Extrasynaptic GABAA receptors: Their function in the CNS and implications for disease. *Neuron* **73**, 23-34 (2012).
9. Donegan J. J., Boley A. M., Yamaguchi J., Toney G. M. & Lodge D. J. Modulation of extrasynaptic GABAA alpha 5 receptors in the ventral hippocampus normalizes physiological and behavioral deficits in a circuit specific manner. *Nat Commun* **10**, 2819 (2019).
10. Brandon NJ, Delmas P, Hill J, Smart TG, Moss SJ. Constitutive tyrosine phosphorylation of the GABA(A) receptor gamma 2 subunit in rat brain. *Neuropharmacology* **41**, 745-52 (2001).
11. Jurd, R., Tretter, V., Walker, J., Brandon, N.J. & Moss, S.J. Fyn kinase contributes to tyrosine phosphorylation of the GABA(A) receptor gamma2 subunit. *Mol Cell Neurosci* **44**, 129-34 (2010).
12. Wells, A.M. et al. Contribution of an SFK-Mediated Signaling Pathway in the Dorsal Hippocampus to Cocaine-Memory Reconsolidation in Rats. *Neuropsychopharmacology* **41**, 675-85 (2016).
13. Bridi, M.C.D. et al. Daily Oscillation of the Excitation-Inhibition Balance in Visual Cortical Circuits. *Neuron* **105**, 621-629 e4 (2020).
14. Khan, M.M. & Regehr, W.G. Loss of Doc2b does not influence transmission at Purkinje cell to deep nuclei synapses under physiological conditions. *Elife* **9**(2020).

REVIEWERS' COMMENTS

Reviewer #1 (Remarks to the Author):

The authors have addressed all of my concerns.

Reviewer #2 (Remarks to the Author):

The authors have responded very positively and extensively to all the criticisms raised against their previous manuscript. Clearly, they have dedicated a great deal of time and energy improving the study. As a result, control data is much improved, more overlapping techniques have been included, and there is now better interpretation of new and existing data. Consequently, the thesis of the paper is much more plausible and should appeal to many in this field. I have just one more question outlined below:

1. In the new data (Fig. 4c), the authors report an increase in efficacy in the presence of GABARAP in new dose response curves. I'm not sure this is interpreted correctly. I think what they are seeing simply mirrors the increase in receptor number at the cell membrane surface in the presence of GABARAP. So it concurs with the biotinylation and increases in current density they report in other plots. The DR curves +/- GABARAP are presumably taken from independent cells and transfections, so normalising to the maximum of the WT data makes no sense, as it misleads the reader into thinking this is an efficacy increase.

Reviewer #3 (Remarks to the Author):

The revised manuscripts addresses my original critiques and now I support publication in Nature Communications.

Point-by-point responses to the comments raised by the reviewers

Before point-by-point responses to the referees' comments, we thank all the reviewers for recognizing the novelty and interests of our works and their critical and constructive suggestions and guidance that help us to efficiently improve our manuscript. Our responses are shown in blue.

Reviewer #1 (Remarks to the Author):

The authors have addressed all of my concerns.

We thank the reviewer for the support.

Reviewer #2 (Remarks to the Author):

The authors have responded very positively and extensively to all the criticisms raised against their previous manuscript. Clearly, they have dedicated a great deal of time and energy improving the study. As a result, control data is much improved, more overlapping techniques have been included, and there is now better interpretation of new and existing data. Consequently, the thesis of the paper is much more plausible and should appeal to many in this field. I have just one more question outlined below:

1. In the new data (Fig. 4c), the authors report an increase in efficacy in the presence of GABARAP in new dose response curves. I'm not sure this is interpreted correctly. I think what they are seeing simply mirrors the increase in receptor number at the cell membrane surface in the presence of GABARAP. So it concurs with the biotinylation and increases in current density they report in other plots. The DR curves +/- GABARAP are presumably taken from independent cells and transfections, so normalising to the maximum of the WT data makes no sense, as it misleads the reader into thinking this is an efficacy increase.

We thank the reviewer for this helpful suggestion. We have now changed the presentation of dose response curve in Fig. 4c.

Fig. 4c: Dose-response curves of I_{GABA} in HEK-293 cells co-expressing GABA_ARs ($\alpha 1\beta 2\gamma 2$) and GABARAP. The data were normalized to I_{max} of each group. GABA at 1 μ M, 3 μ M, 10 μ M, 30 μ M, 100 μ M, 300 μ M and 1000 μ M were selected. Data were fit using the Hill equation with variable slope. Data are represented as the mean \pm SEM. n=7.

Reviewer #3 (Remarks to the Author):

The revised manuscripts addresses my original critiques and now I support publication in Nature Communications.

We thank the reviewer for the support.